# Plant-and-Steal: Truthful Fair Allocations via Predictions

**Ilan Reuven Cohen**
Bar-Ilan University
ilan-reuven.cohen@biu.ac.il

**Alon Eden**
The Hebrew University
alon.eden@mail.huji.ac.il

**Talya Eden**
Bar-Ilan University
talyaa01@gmail.com

**Arsen Vasilyan**
UC Berkeley
arsen@berkeley.edu

## Abstract

We study truthful mechanisms for approximating the Maximin-Share (MMS) value of agents with additive valuations for indivisible goods. Algorithmically, constant factor approximations exist for the problem for any number of agents. When adding incentives to the mix, a jarring result by Amanatidis, Birmpas, Christodoulou, and Markakis [EC 2017] shows that the best possible approximation for two agents and $m$ items is $\lfloor \frac{m}{2} \rfloor$. We adopt a learning-augmented framework to investigate what is possible when a prediction on the input is given. For two agents, we give a truthful mechanism that takes agents' ordering over items as prediction. When the prediction is accurate, our mechanism gives a 2-approximation to the MMS (consistency), and when the prediction is off, our mechanism still obtains an $\lceil \frac{m}{2} \rceil$-approximation to the MMS (robustness). We further show that the mechanism's performance degrades gracefully in the number of "mistakes" in the prediction; i.e., we interpolate between the two extremes: when there are no mistakes, and when there is a maximum number of mistakes. We also show an impossibility result on the obtainable consistency for mechanisms with finite robustness. For the general case of $n \geq 2$ agents, we give a 2-approximation mechanism for accurate predictions, with relaxed fallback guarantees. Finally, we give experimental results which illustrate when different components of our framework, made to ensure consistency and robustness, come into play.

## 1 Introduction

Allocating items among self interested agents in a "fair" way is an age-old problem, with many applications such as splitting inheritance and allocating courses to students. As a starting point, consider the case of two agents. When the items are divisible, the famous cut-and-choose procedure achieves fairness in two senses. Firstly, no agent wants to switch their allocation with the other; i.e., there is no envy among the agents. Secondly, each agent gets a bundle of items which they value at least as much as their value for all the items divided by 2; that is, each one gets their "fair share". When moving to the case of indivisible goods, which is relevant to scenarios such as splitting inheritance and allocating courses, things get trickier. For instance, if there's a single item, the agent that does not receive that item does not get an envy-free allocation, nor do they get their "fair share" according to the previous definitions. Therefore, it is clear that some fairness needs to be sacrificed in this case.

The study of fair allocations with indivisible goods has been a fruitful research direction, with many meaningful notions of fairness studied (see survey by Amanatidis et al. [10]). In this paper, we

38th Conference on Neural Information Processing Systems (NeurIPS 2024).

focus on the notion of the Maximin Share, or MMS, introduced by Budish [18]. For two agents, this notion captures the value an agent will ensure if we implement the cut-and-choose procedure. That is, assume Alice splits the items into two bundles, and then Bob takes one of them (adversarially), and Alice gets the second one. The MMS captures exactly how much value Alice can guarantee for herself. Generalizing the notion for $n$ agents is pretty straightforward — the MMS is the minimum value Alice can guarantee for herself when she partitions the items into $n$ bundles, assuming $n - 1$ bundles are taken adversarially.

We study the case where agents have additive valuations over goods.[1] For the case of two agents, the allocation produced by the cut-and-choose procedure guarantees each of the agents their MMS value. For more than two agents, the existence of such an allocation is not longer guaranteed. Kurokawa et al. [30] show an instance of three agents, where in every allocation, at least one of the agents does not get their MMS value. Since allocating all the agents their MMS value is not always feasible, various papers studied the existence of approximately optimal allocation. An allocation is an $\alpha$-approximate MMS allocation for $\alpha > 1$ if every agents gets at least an $1/\alpha$ fraction of their MMS value. Feige et al. [22] introduce an instance where one cannot find an $\alpha$-approximate allocation for $\alpha < \frac{40}{39}$. On the other hand, [30] show there always exists $\frac{3}{2}$-approximation. The $\frac{3}{2}$ factor was gradually improved [16, 24, 23, 8, 4, 3, 5], where the state-of-the-art algorithm achieves an approximation of $959/720 < 4/3$ [3]. Adding incentives to the mix further complicates matters.

Amanatidis et al. [7] study the case of two additive agents, and $m$ items, where the algorithm (or mechanism) does not know the values of the agents. Thus, the algorithm's designer is faced with the task of devising an allocation rule such that *(i)* agents will maximize their allocated value by bidding truthfully, and *(ii)* the resulting allocation is an $\alpha$-approximate MMS allocation for an $\alpha$ close to 1 as possible. [7] show that no incentive-compatible algorithm can approximate the MMS to a factor better than $\lfloor \frac{m}{2} \rfloor$, and this is matched by the following trivial mechanism — the first agent picks their favorite item, and the second agent gets the rest.

For $2 < n < m$,[2] a trivial truthful algorithm that lets the first $n - 1$ agents pick a single item in some order and gives the last agent the rest achieves an $\lfloor \frac{m-n+2}{2} \rfloor$-approximation, and no better mechanism is known. It is conjectured that one cannot drop the dependence in $m$ for $n > 2$. We are left with a stark disparity. On the one hand, assuming agents' values are public information, approximate solutions are known to exist. On the other hand, when considering private values, it seems that only trivial approximations are possible. *The goal of this paper is to bridge these two regimes using predictions.*

We study the problem of truthful allocations that approximate the MMS, taking a learning-augmented point of view. In the learning-augmented framework, the algorithm designer aims to tackle some intrinsic hardness of the problem at hand, which might arise due to computational constraints, space constraints, input arriving piecemeal online, or incentive constraints, among others. To help the designer overcome these constraints, the algorithm is given some side information which is a function of the input, or a *prediction*, in order to improve the algorithm's performance. The hope is that if the prediction is accurate, then the performance is greatly improved over the performance without the prediction (termed *consistency*). On the other end, if the prediction is inaccurate then the performance of the algorithm is comparable to the performance of the best algorithm that is not given access to predictions (termed *robustness*). The learning-augmented framework has proven useful in bypassing impossibilities that arise due to incentive issues [14, 1, 25, 15, 40, 33, 13].

When designing a learning-augmented mechanism, one should think of realistic predictions. For instance, predicting the entire valuation profile of all agents seems to be a strong assumption. A more plausible assumption is to have some ordinal ranking over the items of the agents. Indeed, it seems unlikely that the algorithm can accurately predict Alice's value for a car, but it is plausible that the algorithm can guess that Alice values the car more than she values the table. Ideally, the algorithm's performance should remain robust if the predicted ordering is almost perfect, with only a few pairs of items whose real ordering is swapped in the prediction. Another desired property is to make the prediction as space-efficient as possible, as previous results [20, 31, 32] show that succinct predictions are crucial for learning parameters from few samples and for incorporating a PAC-learnable component in the learning-augmented framework.

---

[1]Agent $i$ with an additive valuation has a value $v_{ij} = v_i(j)$ for every item, and their value for bundle $S$ is $v_i(S) = \sum_{j \in S} v_{ij}$.

[2]For $n > m$, the MMS of each agent is trivially 0. The problem becomes more interesting for $m \gg n$.

In this paper we devise learning-augmented truthful mechanisms for the problem of approximate-MMS allocations, while taking into considerations the concerns mentioned above.

**Our Results.**   We start by studying the two agent case. We aim at getting: $(a)$ *Constant consistency:* when the predictions are accurate, we want to get a constant approximation to the MMS. $(b)$ *Near-optimal robustness:* when the predictions are off, we want to get as close as possible to the optimal $\lfloor \frac{m}{2} \rfloor$-approximation we can obtain by truthful mechanisms [7].

`Plant-and-Steal` **Framework.**   In Section 3 we present a framework for devising learning-augmented mechanisms for approximating the MMS with two agents. As using only predictions does not guarantee any robustness, we use reports to ensure each agent gets at least one valuable item. This is done while maintaining a near-optimal allocation according to predictions. Our framework, which we term `Plant-and-Steal`, is modular. Along with the set of goods and the agents' reports, it also receives a prediction and an allocation procedure. Different combinations of predictions and allocation procedures yields different consistency-robustness tradeoffs. It is worth noting that, although privacy is not the primary focus of this paper, the `Plant-and-Steal` framework uses agents' reports in a minimal way, as they are only required to select (i.e. "steal back") a single item from a predefined set of options, where this set is determined by the predictions, and not the actual reports.

**Ordering Predictions.**   In Section 4, we study learning-augmented mechanisms when the predictions given are *preference orders over items* of the agents, rather than the values. We instantiate the `Plant-and-Steal` framework with a Round-Robin-based allocation procedure. We observe that in the case of two agents, Round-Robin obtains 2-approximation to the MMS. The 2-consistency of using `Plant-and-Steal` with Round-Robin as the allocation procedure almost immediately follows. The $\lceil \frac{m}{2} \rceil$ robustness follows two facts: $(a)$ Round-Robin produces allocations that are balanced in the number of items allocated to each agent; and $(b)$ The `Plant-and-Steal` framework ensures each agent gets one of their 2 favorite items *according to reports*. In Appendix E, we show how to get an improved $\frac{3}{2}$ consistency, while maintaining $O(m)$ robustness when using a modified Round-Robin allocation procedure.

In Sec 5, we then study the performance of the `Plant-and-Steal` framework when using the Round-Robin procedure, when the prediction given is not fully accurate, but accurate to some degree. To quantify the prediction's accuracy, we adopt the Kendall tau distance measure. The Kendall tau distance counts the number of pairs of elements swapped in the predicted preference order and the order induced by the true valuations. We show that combining the `Plant-and-Steal` framework with a Round-Robin allocation procedure obtains $O(\sqrt{d})$-approximation to the MMS when the Kendall tau distance is $d$. Since $d$ goes from 0 to $\binom{m}{2} = \Theta(m^2)$, we recover the constant consistency when there are no errors, and the $O(m)$ robustness when the number of errors is maximal.

**General Predictions.**   In Appendix G, we study the two-agent case where the mechanism is given access to predictions which are not necessarily the preference order of the agents. We first show that for any prediction given to the learning-augmented mechanism, no mechanism can simultaneously be $\alpha$-consistent while maintaining finite robustness for $\alpha < 6/5$. For the proof, we leverage the characterization of two-agent truthful mechanisms by [7].

We then study small-space predictions. The Round-Robin-based mechanisms described above require an $\Omega(m)$-bit prediction (to describe an arbitrary allocation of items). We first notice that we can implement a bag-filling type allocation procedure using $O(\log m)$-bit predictions. This already achieves a constant consistency along with $O(m)$ robustness. We then devise a more refined allocation procedure, which requires $O(\log m/\epsilon)$-bit predictions, and achieves $2 + \epsilon$ consistency along with $\lceil \frac{m}{2} \rceil$ robustness.

**General number of agents n.**   In Appendix H, we devise a learning-augmented truthful mechanism for $n \geq 2$ additive agents. We obtain a 2-consistent mechanism, while relaxing the robustness guarantees of the mechanism. We take a similar approach to the work of [18, 27, 28, 2, 5], who compete against a relaxed benchmark of the MMS value for $\hat{n} > n$ agents, and try to minimize $\hat{n}$. We obtain an $\max\{m - \hat{n} - 1, 1\}$-approximation to the MMS for $\hat{n} = \lceil \frac{3n}{2} \rceil$ agents when the predictions are off. Our mechanism uses the modified Round-Robin procedure from [8] to determine the initial

allocation using the predictions. It then applies a recursive plant-and-steal procedure to determine the final allocation.

**Experiments.** Finally, In Section 6, we demonstrate how several components in our design come into play when experimenting with synthetic data. We run different variants of mechanism on two player instances, and show that when predictions are accurate, then only using predictions is nearly optimal, if predictions are noisy, then the stealing component ensures robustness, and our `Plant-and-Steal` framework achieves best-of-both-worlds guarantees.

We summarize the bounds we obtain in Table 1.

| Setting | Consistency | Robustness | Reference |
|---|---|---|---|
| Ordering predictions, $n = 2$ | 2 | $\lceil m/2 \rceil$ | Section 4 |
| | 3/2 | $\lfloor 2m/3 \rfloor$ | Section 4 |
| | $\geq 5/4$ | Any | [6] |
| Arbitrary predictions, $n = 2$ | Any | $\geq \lfloor m/2 \rfloor$ | [7] |
| | $\geq 6/5$ | Bounded | Section G.1 |
| $3 \log m + 1$ space | 4 | $m - 1$ | Section G.2 |
| $O(\log(m)/\epsilon)$ space | $2 + \epsilon$ | $\lceil m/2 \rceil$ | Section G.3 |
| $n > 2$ | 2 | $m - \lceil 3n/2 \rceil - 1$ for $\hat{n} = \lceil 3n/2 \rceil$ | Section H |

Table 1: Known bounds for truthful learning-augmented MMS mechanisms.

**Further Related Work.** In addition to the the studies mentioned above, we give a comprehensive review of further related work in Appendix B.

## 2 Preliminaries

In the setting we study, there is a set $N$ of $n$ agents and a set $M$ of $m$ indivisible items. Each agent has a *private* additive valuation over the items, unknown to the mechanism designer, where the value of agent $i$ for item $j$ is $v_{ij}$ (also denoted as $v_i(j)$). For a bundle $S \subseteq M$ of items, $v_i(S) = \sum_{j \in S} v_{ij}$.

The fairness notion we focus on is the following.

**Definition 2.1** (Maximin Share). *The Maximin Share (MMS) of agent $i$ with valuation $v_i$ and $n$ agents is*

$$\mu_i^n = \max_{S_1 \bigcup \ldots \bigcup S_n = M} \min_{j \in [n]} v_i(S_j);$$

*that is, if $i$ were to partition the items into $n$ bundles, and then $n - 1$ of those bundles are taken adversarially, what is the value $i$ can guarantee for themselves. When clear from the context, we omit $n$ and use $\mu_i$ to denote the MMS of $i$ with $n$ agents.*

We are interested in mechanisms that produce approximately optimal allocations, as defined next.

**Definition 2.2** (($\gamma, k$)-approximate MMS Allocation). *An allocation $X = (X_1, \ldots, X_n)$ is $(\gamma, k)$-approximate MMS allocation for $\gamma > 1$ and a natural number $k$ if for every agent $i$,*

$$v_i(X_i) \geq \mu_i^k / \gamma.$$

*When $k = n$, we say the allocation is a $\gamma$-approximate MMS allocation.*

We study mechanism that get some prediction on the input.

**Definition 2.3** (Learning-Augmented Mechanism). *A learning-augmented mechanism takes agents' reports $\mathbf{r} = (r_1, \ldots, r_n)$ and predictions $\mathbf{p}$ in some prediction space $\mathcal{P}$, and outputs a partition of the items*

$$X(\mathbf{r}, \mathbf{p}) = (X_1(\mathbf{r}, \mathbf{p}), X_2(\mathbf{r}, \mathbf{p}), \ldots, X_n(\mathbf{r}, \mathbf{p})), \quad X_1(\mathbf{r}, \mathbf{p}) \bigcup X_2(\mathbf{r}, \mathbf{p}) \bigcup \ldots \bigcup X_n(\mathbf{r}, \mathbf{p}) = M,$$

*where agent $i$ gets $X_i(\mathbf{r}, \mathbf{p})$.*

For learning-augmented mechanisms, truthfulness should hold for any possible prediction $\mathbf{p}$.

**Definition 2.4.** *A learning-augmented mechanism is truthful if for every agent $i$ and every possible report of other agents $\mathbf{r}_{-i}$ and every possible prediction $\mathbf{p}$,*

$$v_i(X_i(v_i, \mathbf{r}_{-i}, \mathbf{p})) \geq v_i(X_i(r_i, \mathbf{r}_{-i}, \mathbf{p}))$$

*for every $r_i$.*

We next define the consistency and robustness measures according to which we measure the performance of our mechanisms.

**Definition 2.5** ($\alpha$-consistency). *Consider a prediction function $f_{\mathcal{P}}$ which takes a valuation profile and outputs a prediction in prediction space $\mathcal{P}$. A learning-augmented mechanism is $\alpha$-consistent for $\alpha > 1$ and prediction function $f_{\mathcal{P}}$ if for every valuation profile $\mathbf{v}$ and every prediction $\mathbf{p} = f_{\mathcal{P}}(\mathbf{v})$, $X(\mathbf{v}, \mathbf{p})$ is an $\alpha$-approximate MMS allocation.*

**Definition 2.6** (($\beta, k$)-robust). *A learning-augmented mechanism is $(\beta, k)$-robust for $\beta > 1$ and natural number $k$ if for every valuation profile $\mathbf{v}$ and every prediction $\mathbf{p}$, $X(\mathbf{v}, \mathbf{p})$ is an $(\beta, k)$-approximate MMS allocation. If $k = n$, we say the mechanism is $\beta$-robust.*

For ease of presentation, for valuation $v_i$, report $r_i$ and prediction $p_i$, we use $v_i^\ell, r_i^\ell, p_i^\ell$ to denote *both* the $\ell^{\text{th}}$ highest good according to the valuation/report/prediction *and* its value. Note that, we may use $v_i^\ell$ for $\ell > m$, in this case, $v_i^\ell = 0$. For $\ell = 1$, i.e., the highest good we use $v_i^*, r_i^*, p_i^*$.

**Ordering Predictions and Kendall tau Distance.**    Most of our mechanisms use predictions which take the form of an ordering over agents' items. That is, $f_{\mathcal{P}}(\mathbf{v})$ outputs a vector of orderings $\mathbf{p} = (p_1, \ldots, p_n)$, where $p_i^\ell$ is the $\ell^{\text{th}}$ highest valued item of $i$ in $M$ according to $\mathbf{p}$. Accordingly, for agent $i$, let $v_i^\ell$ be the $\ell^{\text{th}}$ highest valued item according to $\mathbf{v}$. For two items $j \neq j'$, We use $j \succ_{p_i} j'$ to denote that $j$ is higher ranked than $j'$ according to $\mathbf{p}$.

When studying imprecise predictions, we want to quantify the degree to which the prediction is inaccurate. For this, we use the following measure. For an agent $i$, we define our noise level with respect to the Kendall tau distance (also known as bubble-sort distance) between $\mathbf{v}$ and $\mathbf{p}$.

**Definition 2.7** (Kendall tau distance). *The Kendall tau distance counts the number of pairwise disagreements between two orders. For $i$'s valuation $v_i$ and predicted preference order $p_i$, we define*

$$K_d(v_i, p_i) = |\{j \succ_{p_i} j' \; : \; v_i(j) < v_i(j')\}|.$$

*That is, the number of pairs of items where the prediction got their relative ordering wrong. We also denote $K_d(\mathbf{v}, \mathbf{p}) = \max\{K_d(v_1, p_1), K_d(v_2, p_2)\}$.*

We note that the Kendall tau distance between $v_i$ and $p_i$, $K_d(v_i, p_i)$, can go from 0 to $\binom{m}{2}$.

## 3   `Plant-and-Steal` **Framework**

In this section, we present the framework which is used to devise learning-augmented mechanisms for two agents. The ideas presented here also inspire the more complex learning-augmented mechanism for $n > 2$ agents. Missing proofs of this section appear in Appendix C. Our framework, which we term `Plant-and-Steal` is given the set of goods, an allocation procedure $\mathcal{A}$, the prediction $\mathbf{p}$ and reports $\mathbf{v}$. The framework operates as follows:

1. It first applies $\mathcal{A}$ on the predictions $\mathbf{p}$ to divide the set of goods into two bundles $A_1, A_2$. The procedure $\mathcal{A}$ should be an allocation procedure with good MMS guarantees. We use different allocation procedures depending on the type of prediction given and on the consistency-robustness tradeoffs we are aiming for.
2. *Planting phase:* For each agent $i$, it picks $i$'s favorite item in set $A_i$ *according to prediction*, and "plants" this item in the bundle $A_j$ of the other agent $j \neq i$. Let $T_1, T_2$ denote the sets that result in this planting phase.
3. *Stealing phase:* To obtain the final allocation, each agent $i$ now "steals" back their favorite item from set $T_j$ of agent $j \neq i$ *according to reports*. Notice this is the first and only place where we use agents' reports.

This procedure is trivially truthful because the only step where we use agents' reports is the one where they pick exactly one item to steal back from $T_j$, and this $T_j$ only depends on predictions, and not reports (Lemma 3.1). To obtain robustness, we notice that each agent gets one of their two favorite items according to their true valuations (Lemma 3.2). This implies a robustness of $m-1$. We show that for balanced allocations, we get improved robustness guarantees (Lemma 3.4).

For $S \subseteq M$, and agent $i$, let $v_i^*(S)$ ($p_i^*(S), r_i^*(S)$) be the max valued item in $S$ according to $v_i$ ($p_i, r_i$). for $g \in M$ and $S \subseteq M$, denote $S + g := S \cup \{g\}$ and $S - g = S \setminus \{g\}$. The `Plant-and-Steal` framework is presented in Mechanism 1.

---

**MECHANISM 1:** Two agent `Plant-and-Steal` Framework

**Input** : Allocation Procedure $\mathcal{A}$, set of items $M$, predictions $\mathbf{p}$ and reports $\mathbf{r}$
**Output:** Allocations $X_1 \bigcup X_2 = M$

                    /* Find an initial allocation by applying $\mathcal{A}$ on the predictions */
$(A_1, A_2) := \mathcal{A}(M, N, \mathbf{p})$

                    /* Plant favorite items according to predictions */
$\hat{j}_1 \leftarrow p_1^*(A_1)$
$\hat{j}_2 \leftarrow p_2^*(A_2)$
$T_1 \leftarrow A_1 + \hat{j}_2 - \hat{j}_1$
$T_2 \leftarrow A_2 + \hat{j}_1 - \hat{j}_2$

                    /* Steal according to report */
$\tilde{j}_1 \leftarrow r_1^*(T_2)$
$\tilde{j}_2 \leftarrow r_2^*(T_1)$
$X_1 \leftarrow T_1 + \tilde{j}_1 - \tilde{j}_2$
$X_2 \leftarrow T_2 + \tilde{j}_2 - \tilde{j}_1$

---

We show that for any allocation function $\mathcal{A}$ and predictions $\mathbf{p}$ given to the framework, the resulting mechanism is truthful.

**Lemma 3.1** (Truthfulness Lemma). *For any allocation procedure $\mathcal{A}$, `Plant-and-Steal` mechanism using $\mathcal{A}$ is truthful.*

Since the framework is truthful, from now on, we assume that $\mathbf{r} = \mathbf{v}$. Next, we show that the `Plant-and-Steal` mechanism ensures that for each agent, an item is allocated with a value that is at least as good as their second-best option *according to their value*.

**Lemma 3.2.** *Consider the allocation $(X_1, X_2)$ returned by `Plant-and-Steal` with some allocation procedure $\mathcal{A}$. For any agent $i$, then $v_i^1 \in X_i$ or $v_i^2 \in X_i$.*

We next claim that if $i$ gets one of their two favorite items and any $k-1$ additional items, $i$'s value is an $m-k$-approximation to $\mu_i$.

**Lemma 3.3.** *For any agent $i$, let $S \subseteq M$ be a subset of the items of size $|S| = k$ and $v_i^1 \in S$ or $v_i^2 \in S$ then*

$$v_i(S) \geq \mu_i/(m-k).$$

We immediately get the following.

**Lemma 3.4** (Robustness Lemma). *Let $\mathcal{A}$ be an allocation rule guaranteeing $\min\{|A_1|, |A_2|\} \geq k$, then when `Plant-and-Steal` uses $\mathcal{A}$, the resulting mechanism is $(m-k)$-robust.*

*Proof.* By Lemma 3.2, we are guaranteed that each agent gets one of their two favorite items according to their report. Combining with the condition on $\mathcal{A}$ and Lemma 3.3, the proof is finished. □

## 4 Ordering Predictions

In this section, we consider the case of two agents, where the predictions (and in fact, also the reports) given to the mechanism are preference orders of agents over items. Our mechanisms makes use of the `Plant-and-Steal` framework instantiated by Round-Robin based allocation procedures. We

first present the round-robin allocation procedures we'll use, and give their approximation guarantees when the input is accurate. Next, we prove the robustness and consistency guarantees. Finally, we quantify the accuracy of the predictions using the Kendall tau distance, and obtain fine-grained approximation results, where the approximation smoothly degrades in the accuracy.

Amanatidis et al. [6] studied mechanisms where the preference orders of the agents over items are public (while valuations are private). They showed that no truthful mechanism can achieve a better approximation than $5/4$ in this setting. This implies that when the predictions are preference orders, no learning-augmented mechanism can obtain consistency better than $5/4$, no matter if the robustness is bounded or not.

**Proposition 4.1** (Corollary of Amanatidis et al. [6]). *For any $\epsilon > 0$, no mechanism that is given preference orders as predictions can obtain consistency $5/4 - \epsilon$.*

**Round-Robin Allocation Procedures.** The two allocation procedures we use to instantiate the `Plant-and-Steal` framework take as input preference orders of agents over items:

- `Balanced-Round-Robin`: the agents take turns, and at each turn, an agent takes their highest ranked remaining item. This results in a balanced allocation.
- `1-2-Round-Robin`: the agents take turns, where we compensate the second agent, who might not get their favorite item, to take two items each turn.

In this section, we only prove consistency-robustness guarantees when `Balanced-Round-Robin` is used as the allocation procedure. In Appendix E we show different tradeoffs when `1-2-Round-Robin` is used.

---

**ALGORITHM 2:** `Balanced-Round-Robin`

**Input** : Preference orders of agents over items $\mathbf{v} = (v_1, v_2)$.
**Output** : An allocation $A_1 \bigcup A_2 = M$.
$A_i \leftarrow \emptyset$ for every agent $i \in \{1, 2\}$
**for** $r = 1, \ldots, \lceil |M|/2 \rceil$ **do**
$\quad A_1 \leftarrow A_1 + v_1^*(M \setminus A_1 \setminus A_2)$
$\quad A_2 \leftarrow A_2 + v_2^*(M \setminus A_1 \setminus A_2)$

---

Consider the allocation procedure depicted in Algorithm 2. In order to implement the two allocation procedures, we only needs to receive preference orders over items. Let $A_i = (a_i^1, \ldots, a_i^{|A_i|})$ be agent $i$'s allocation by the algorithm, where $a_i^k$ is the $k$'th choice of agent $i$. We observe the following.

**Observation 4.1.** *The output $(A_1, A_2)$ of the `Balanced-Round-Robin` procedure, satisfies:*

1. *$|A_1| = \lceil \frac{m}{2} \rceil$, $|A_2| = \lfloor \frac{m}{2} \rfloor$.*
2. *For each agent $i$ and round $k$, $a_i^k \in \{v_i^\ell\}_{\ell \in [2k]}$; that is, in round $k$ an agent gets one of their top $2k$ items.*

Amanatidis et al. [8] show that first allocating large items to agents, and then using a Round-Robin to allocate the remaining items to the remaining agents, gives a 2-approximation to the MMS. We observe that for two agents, Round-Robin *as is*, without the initial step, achieves this approximation guarantee. The proof of the following Lemma is deferred to Appendix D.

**Lemma 4.1.** *Let $(A_1, A_2)$ be the allocation of `Balanced-Round-Robin`. For every agent $i$, $v_i(A_i) \geq \mu_i/2$.*

We next use the allocation procedure to instantiate the `Plant-and-Steal` framework.

**Round-Robin-Based Mechanism.** The mechanism we analyze, `B-RR-Plant-and-Steal`, results from instantiating `Plant-and-Steal` with `Balanced-Round-Robin` as $\mathcal{A}$.

We first show that if the predictions correspond to the preference orders of the real valuations, then `B-RR-Plant-and-Steal` outputs the same allocation as `Balanced-Round-Robin`.

**Lemma 4.2.** *When predictions correspond to actual values, `B-RR-Plant-and-Steal` outputs the same allocation as `Balanced-Round-Robin`.*

We are now ready to prove the performance guarantees of our mechanisms.

**Theorem 4.1.** *Mechanism* `B-RR-Plant-and-Steal` *is truthful,* 2*-consistent and* $\lceil \frac{m}{2} \rceil$*-robust.*

*Proof.* By Lemma 3.1, the mechanism is truthful. By Observation 4.1, each agent receives at least $\lfloor m/2 \rfloor$ items; combining with Lemma 3.4, we get that the mechanism is $\lceil \frac{m}{2} \rceil$-robust. Finally, if predictions correspond to valuations, by Lemma 4.1 and Lemma 4.2, the allocation is a 2-approximation to the MMS. Thus, the mechanism is 2-consistent. $\square$

We note that by Amanatidis et al. [7], our robustness guarantee matches the optimal obtainable approximation by any truthful mechanism (up to the rounding).

## 5 Noisy Predictions

We now analyze Mechanism `B-RR-Plant-and-Steal`'s performance under varying levels of noise. Consider the case where the Kendall tau distance between $\mathbf{v}$ and $\mathbf{p}$ is at most $d$. Our main theorem in this section shows that combining the `Plant-and-Steal` framework with a Round-Robin allocation procedure obtains $O(\sqrt{d})$-approximation to the MMS when the Kendall tau distance is $d$. Missing proofs of this section appear in Appendix F.

To prove the approximation ratio, we relate the value that agent $i$ obtains from the allocation, $v_i(X_i)$, to their maximin share, $\mu_i$, by considering the worst possible set of items that agent $i$ might receive under the Round-Robin procedure when acting on their true preferences. Specifically, we define this worst-case set as $R_i = \{v_i^{2j}\}_{j \in \{1, \ldots, \lfloor m/2 \rfloor\}}$. In Eq. (2) of Lemma 4.1, we prove that $v_i(R_i) \geq \mu_i/2$. Therefore, obtaining an allocation that is a factor of $c$ times $v_i(R_i)$ ensures a factor of $c/2$ of the MMS value.

We further simplify the analysis by applying *the zero-one principle*[3]. The zero-one principle basically let's us reduce the analysis to instances where the values are either 0's or 1's. For threshold $\tau \geq 0$, let $h_\tau(q) = 1$ if $q \geq \tau$ and 0 otherwise, and let $v_i^\tau(S) = \sum_{j \in S} h_\tau(v_i(j))$.

By the zero-one principle, for two sets $S, T \subseteq M$, in order to show that $v_i(S)$ approximates $v_i(T)$, it is enough to show that $v_i^\tau(S)$ approximates $v_i^\tau(T)$ for every threshold $\tau \geq 0$.

**Lemma 5.1.** *For $c > 1$ and for any two sets $S, T \subseteq M$, if for every threshold $\tau \geq 0$, $v_i^\tau(S) \geq v_i^\tau(T)/c$, then $v_i(S) \geq v_i(T)/c$.*

Thus, we will show that when the Kendall tau distance is $d$, for every threshold $\tau \geq 0$, $v_i^\tau(X_i) \geq v_i^\tau(R_i)/c$ for some $c = O(\sqrt{d})$. Recall that $A_i$ is the set of items assigned to $i$ after running the Round-Robin procedure on the predictions $\mathbf{p}$. We first show that for Kendall tau distance $d$, the *additive* approximation $v_i^\tau(A_i)$ gives to $v_i^\tau(R_i)$ is $\sqrt{d}$.

**Lemma 5.2.** *If the Kendall tau distance between $\mathbf{p}$ and $\mathbf{v}$ is at most $d$, then for any threshold $\tau \geq 0$, we have that $v_i^\tau(A_i) \geq v_i^\tau(R_i) - \sqrt{d}$.*

We note that although $v_i^\tau(A_i)$ gives an additive approximation to $v_i^\tau(R_i)$, it can still be the case that the Kendall tau distance is constant, yet $v_i(A_i)$ does not give any multiplicative approximation to $\mu_i$.[4] Therefore, we must use the fact that agent $i$ gets to "steal" an item according to their *true* valuation in the `Plant-and-Steal` procedure in order to get our approximation guarantee. By combining these results, we prove the following theorem concerning the approximation ratio of `B-RR-Plant-and-Steal`'s performance under varying levels of noise, $d$. [5]

**Theorem 5.1.** *Consider a prediction $\mathbf{p}$ and valuations $\mathbf{v}$ such that $K_d(\mathbf{v}, \mathbf{p}) = d$, then Mechanism* `B-RR-Plant-and-Steal` *gives a* $(2\sqrt{d} + 6)$*-approximation to the MMS.*

---

[3]Applied in [11], for instance, in the context of packet routing.

[4]Indeed, consider the case where there are four goods which both agents value at $(1, 1, 0, 0)$. If agent $i$'s prediction orders the last two items higher then the first two items, we will get that $v_i(A_i) = 0$, while $\mu_i = 1$.

[5]A similar analysis for Mechanism `1-2-RR-Plant-and-Steal` will show a similar dependence on $\sqrt{d}$ (up to constant factors).

## 6   Experimental Results

In this section[6], we give experiments which illustrate the role of different components of our framework for two players under various noise levels of the predictions. The predictions we use for our experiments are the predicted values of the items. The noise we introduce permutes the vectors of values to match the instance's Kendall tau distance, and uses the permuted vector as prediction. We show that our framework is almost optimal for small amounts of noise while still showing resilience for higher noise levels. Moreover, we study the performance of variants which only use specific components of our framework.

When using predictions, our initial allocation procedure is a cut-and-choose procedure, implemented as follows:

- We use the first player's prediction to implement a bag-filling algorithm which sorts the items by values, and then partitions the items into two sets using a greedy procedure that assigns each item to the set with current lowest value.
- We use the second player's prediction to allocate the agent the set with the higher predicted value of the two.

This allocation ensures that the second agent obtains their MMS value according to the prediction. In the data we generates, we observe that in a sampled valuation, the two sets chosen by the bag-filling algorithm gives the two sets the same value, up to $0.5\%$, which ensures that the lowest valued set obtains a $1.026$-approximation to the MMS.

We inspect the following mechanisms:

1. *Random*: a mechanism that ignores reports and predictions and randomly partitions the items into two sets of size $m/2$.
2. *Random-Steal*: a mechanism that ignores predictions, randomly partitions the items into two sets of size $m/2$, and then implements the stealing phase where each player takes their favorite item from the other player's set according to reports.
3. *Partition*: a mechanism that ignores reports, and partitions the items according to predictions, using the cut-and-choose procedure described above.
4. *Partition-Steal*: a mechanism that partitions the items according to predictions, using the cut-and-choose procedure described above, and then implements the stealing phase where each player takes their favorite item from the other player's set according to reports.
5. *Partition-Plant-Steal*: a mechanism that implements the `Plant-and-Steal` framework. partitions the items according to predictions, using the cut-and-choose procedure described above, "plants" each player's favorite item according to predictions, and then "steals" each player's favorite item from the other player's set according to reports.

**Experiments.**   We consider two-player scenarios with $m = 100$ items. For each distance measure, we generate 1000 valuation profiles. For each pair of valuation profiles and corresponding Kendall tau distance, we generate 100 predictions based on the distance. We then assess the performance of the mechanisms described earlier on these instances. We examine two distinct cases regarding the relationship between the players' preference orders: the *Correlated* case, where both players have identical preference orders, although their valuation magnitudes differ, and the *Uncorrelated* case, where the preference orders of the players are generated independently and chosen uniformly at random. Further details on the procedures used to generate the valuations and predictions are provided in Appendix A.

**Benchmark.**   We plot the percentage of these instances where both players get at least $(1 - \epsilon)$ of their MMS value for $\epsilon = 0.1, 0.05, 0.02$.

**Results.**   The results are shown in Figure 1. We first examine the performance of the two mechanisms that do not use predictions, *Random* and *Random-Steal*. Scenarios with correlated values perform significantly worse, as there is a non-negligible probability of an unbalanced partition of the relatively few high and medium valued items in a random partition. For $\epsilon$ values of $0.02, 0.05, 0.1$, the *Random* strategy success rate is $11\%, 25\%$, and $43\%$, respectively, under correlated preferences, compared

---

[6]The experiments, reproducible via Matlab (2022b) at https://tinyurl.com/PlantStealExperiments, were performed on a standard PC (Intel i9, 32GB RAM) in about 30 minutes.

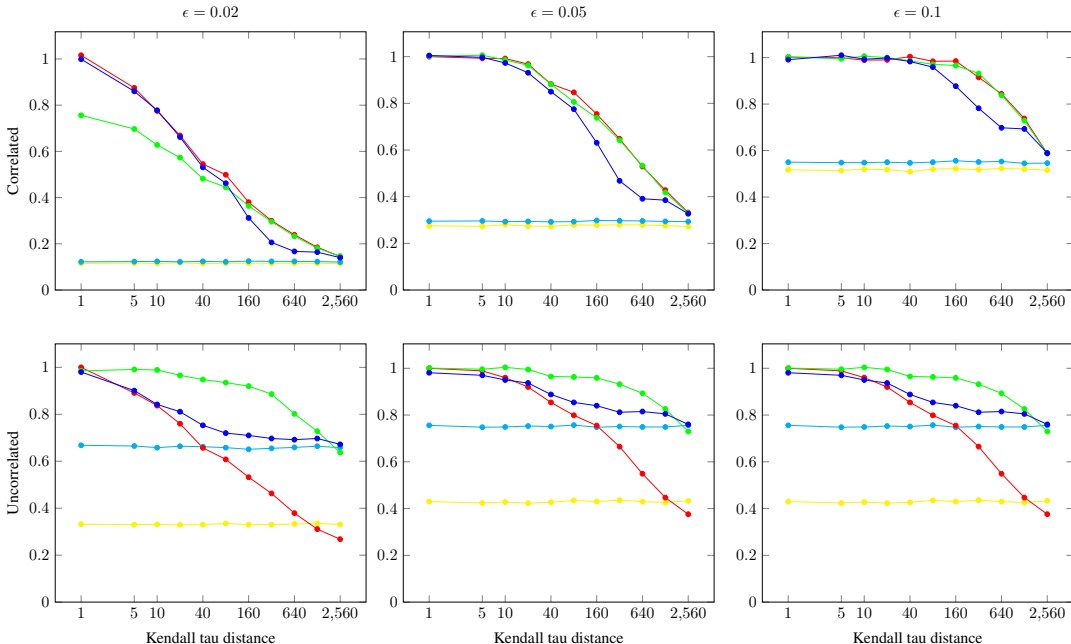

Figure 1: Mechanisms: *Random* (yellow), *Random-Steal* (cyan), *Partition* (red), *Partition-Steal* (green), *Partition-Plant-Steal* (blue). Data generation: correlated (first row) and uncorrelated (second row). Success rate: the percentage of instances where both players receive at least $(1 - \epsilon)$-fraction of their MMS values for different values of $\epsilon$: $0.02$ (first column), $0.05$ (second column), and $0.1$ (third column).

to $33\%, 43\%$, and $60\%$ under uncorrelated preferences. Moreover, adding the stealing component significantly improves the success rate only in the uncorrelated case, as *Random-Steal* achieves success rates of $66\%, 75\%$, and $87\%$. In the correlated case, as each player has a highly valuable item stolen, their obtained value is not expected to increase.

In the mechanisms that use predictions, *Partition*, *Partition-Steal* and *Partition-Plant-Steal*, the performance degrades as a function of noise, as expected. When comparing the performance of *Partition*, which only relies on the prediction component of our framework, and *Random-Steal*, which only relies on the stealing component of our framework, we notice that in the uncorrelated case, for small amount of noise guarantee a higher success rate, while as the noise increases, the stealing component becomes more instrumental to the performance. This is in tact with the theoretical results, where using the prediction is crucial to achieve the consistency guarantees, which take place when the prediction is accurate, while stealing is important to achieve robustness guarantees in case the prediction is inaccurate. As described above, in the case where the valuations are correlated, stealing is not expected to help. Interestingly, on fully noisy input, even *Random* outperforms *Partition* as *Partition* might partition the items into unequally-sized sets, which performs worse than the equally-sized sets *Random* outputs.

Our experiments show that *Partition-Plant-Steal* performs as well as the *Partition* strategy for small amounts of noise and outperforms it on uncorrelated instances for large amounts of noise. Moreover, for any amount of noise, it outperforms *Random-Steal* and converges to it for a fully noisy input. This illustrates the "best of both worlds" tradeoff obtained by our framework.

Finally, when comparing the *Partition-Plant-Steal* strategy to the *Partition-Steal* strategy, we observe that *Partition-Plant-Steal* outperforms *Partition-Steal* in the correlated case with a small amount of noise (worst-case scenario) for $\epsilon = 0.02$, as planting guarantees your favorite items would not be taken. In other scenarios, *Partition-Steal* outperforms *Partition-Plant-Steal* because "planting" removes a valuable item from the player's set that might be taken otherwise, especially in the uncorrelated case.

## Acknowledgments and Disclosure of Funding

Arsen Vasilyan was supported in part by NSF awards CCF-2006664, DMS-2022448, CCF-1565235, CCF-1955217, CCF-2310818, Big George Fellowship and Fintech@CSAIL. The work of A. Vasilyan was partially done while visiting Bar-Ilan university as a part of the MISTI-Israel program, supported by the Zuckerman Institute. Part of this work was conducted while Arsen Vasilyan was visiting the Simons Institute for the Theory of Computing.

The work of I.R. Cohen was supported in part by ISF grant 1737/21. The work of A. Eden was supported by the Israel Science Foundation (grant No. 533/23).

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

## A    Experimental Supplement

**Generating valuations.**    To generate interesting valuations for the players, we use a multi-step function to generate item values, since if the values are close together, any balanced partition obtains good MMS guarantees, without considering reports and predictions. Specifically, we consider a four-step (High/Med/Low/Extra-Low) random valuation function, where an item has a High valuation with probability $8/m$, a Medium valuation with probability $1/4$, a Low valuation with probability $1/2$ and an Extra-Low valuation with the remaining probability. A High valuation is drawn from $U[1000, 2000]$, a Medium valuation is drawn from $U[400, 800]$, a Low valuations is drawn from $U[100, 200]$ and an Extra-Low valuation is sampled from $U[1, 2]$. Figure 2 shows the value distribution generated by this process for two players. We generate values for $m = 100$ items.

We generate valuations satisfying one of the two types of relations between players' preferences:

- *Correlated*: the two preference orders are identical (but not the values).
- *Uncorrelated*: Both preference orders are chosen independently and uniformly at random.

**Generating predictions.**    To generate predictions, we take valuations and permute elements randomly to create noise. We generate predictions under varying noise levels according to the Kendall tau distance between the valuations and the predictions. We very the Kendall tau distance between 1 to 2560, where 2560 corresponds to the expected noise level of a random permutation of 100 items. To randomly choose a permutation of a certain noise level, we start with the ordered permutation and then choose two indices $j < k$ u.a.r. and swap items $r$ and $r + 1$ for $r \in \{j, \ldots, k - 1\}$ if it increases the Kendall Tau distance by one. We repeat this process until the distance of the resulting permutation equals the desired value.

## B    Related Work

The notion of the maximin share allocation was introduced by Budish [18] as an ordinal notion, and extended to the notion we adopt by Bouveret and Lemaître [17]. Using machine learning advice in algorithm design was used in theory [21, 38] and practice [29]. The learning-augmented framework of studying consistency-robustness tradeoffs was introduced by Lykouris and Vassilvitskii [34]. [35, 39] studied the performance of algorithms using imprecise predictions.

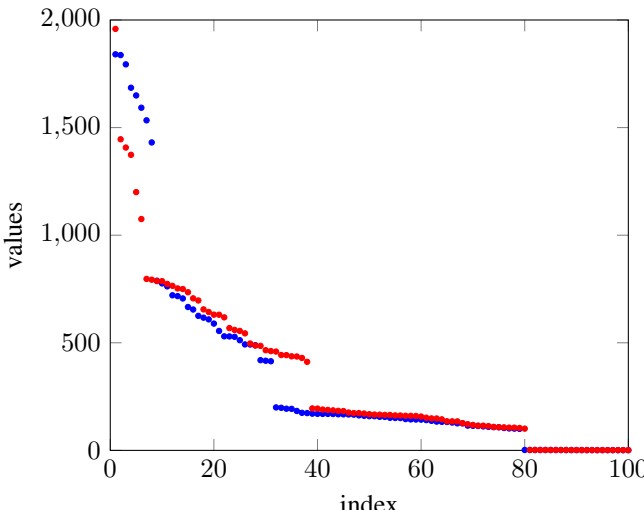

Figure 2: Plotting randomly sampled valuations for two players, where the values are sorted such that lower indexed items have higher values.

**Fair division with incentives.** The two closest papers to ours are Amanatidis et al. [6, 7]. In [6], they initiate the study of truthful mechanisms for approximating the MMS value for agents with additive valuations. They show that no truthful mechanism can get an approximation better than $1/2$ for the MMS in the case of 2 agents and 4 items. They give the best known approximation guarantee for $n$ agents and $m$ items of $\lfloor \frac{m-n+2}{2} \rfloor$. Finally they consider the public ranking model, where the ranking over items is public information. Using this, they are able to obtain a $\frac{n+1}{2}$-approximation algorithm. One can view this as an algorithm that is given a prediction over the input, but does not provide robustness guarantees. [7] Fully characterize truthful mechanism for 2 agents with additive valuations. They use this characterization to provide a strong lower bound of $\lfloor \frac{m}{2} \rfloor$ for any truthful mechanism.

[12] design truthful mechanisms for dichotomous submodular valuations that maximize welfare, along with desirable fairness properties such as EFX and NSW. For additive binary valuations, they also maximize the MMS in a truthful manner. [26] bypass the impossibilities imposed by [7, 37] for truthful fair allocations with indivisible and divisible goods by considering Bayesian Incentive Compatible mechanisms with symmetric priors. They are able to obtain EF-1 allocations for indivisible goods and proportional allocations for indivisible goods.

Finally, [9] study the Nash equilibrium for simple mechanisms for agents with additive valuations. They show that for every number of agents, the Pure Nash equilibrium of the Round-Robin procedure produces an EF-1 allocation. For two agents, they show that the Pure Nash equilibrium of Plaut and Roughgarden [36] cut-and-choose procedure produces an EFX and MMS allocation.

**1-out-of-$k$.** As stated above, the MMS value of an agent is defined by the highest value an agent can guarantee for themselves when partitioning the items into $n$ different bundles, where $n$ is the number of agents, and then getting the lowest valued bundle. Thus, an agent get a value larger than the worst one-out-of-$n$ bundles that define the MMS.

Noticing that finding an allocation that satisfies the MMS value of each agent is a demanding task (which was shown to be infeasible in some cases by Kurokawa et al. [30]), Budish [18] relaxed the notion and defined the 1-out-of-$n+1$ MMS to be the worst bundle out of the bundles that define the MMS when partitioning the items using an additional bundle. [18] showed it is possible to achieve this benchmark when adding a small number of access goods. There has been an effort to find the smallest $k$ for which an allocation that guarantees a 1-out-of-$k$ MMS for each agent exists. [2] were able to show the existence for $k = 2n - 2$, [27, 28] achieved $k = \lceil \frac{3n}{2} \rceil$, and recently, [5] showed the smallest up-to-date $k = \lceil \frac{4n}{3} \rceil$. In our $n$-agent mechanism, our robustness guarantee approximates this relaxed benchmark for $k = \lceil \frac{3n}{2} \rceil$.

**Learning-Augmented Mechanisms.** Agrawal et al. [1] and Xu and Lu [40] first explored the learning-augmented framework in a mechanism design setting, where [1] studied the facility location problem while [40] applied the framework to several settings such as revenue-maximization, path auctions, scheduling and two-facility games. [14] give nearly optimal consistency-robustness tradeoffs to the strategyproof scheduling with unrelated machines. [25] use predictions to design mechanisms with improved Price of Anarchy bounds. [33, 19] study revenue maximization auctions with predictions, and [13] devise bicriteria mechanisms.

## C  Deferred proofs from Section 3

*Proof of Lemma 3.1.* We show that agent 1 is better off reporting their true valuation, a symmetric argument holds for agent 2. First, notice that sets $T_1$ and $T_2$ are determined using predictions, ignoring the reports. Next, notice that the item $\tilde{j}_2$ is chosen only using agent 2's report. Therefore, the only way agent 1 can affect their allocation is by choosing which item in $T_2$ is allocated to them. agent 1 gets their favorite item in $T_2$ according to their report. Therefore, it is clear that the agent maximize their utility by reporting their true value. □

*Proof of Lemma 3.2.* Consider some agent $i$. We claim for every partition of the items into two non-empty sets, $T_1, T_2$, $i$ is always guaranteed to have one of their two favorite items according to their true valuation $v_i$ in $X_i$. This is because either *(1)* $i$ has one of their two favorite items in $T_\ell$, $\ell \neq i$, and $i$ gets their favorite item from $T_\ell$; or $(2)$ $i$'s two favorite items are in $T_i$, and in this case, $i$ gets all items from $T_i$ but one, so $i$ is guaranteed one of them. □

*Proof of Lemma 3.3.* Let $g \in S \cap \{v_i^1, v_i^2\}$, by the definition of $S$ such $g$ exists. Let $S' = S \setminus \{g\}$, by the definition of $S$, we have $|S'| = k - 1$ and $v_i(S) \geq v_i^2 + v_i(S')$. Consider a partition

$$(S_1, S_2) \in \arg\max_{(T_1, T_2)\,:\,T_1 \bigcup T_2 = M} \min_{j \in \{1,2\}} v_i(T_j).$$

By definition, $\mu_i = \min_{j \in \{1,2\}} v_i(S_j)$. We have,

$$
\begin{aligned}
\frac{\mu_i}{v_i(S)} &\leq \frac{\mu_i}{v_i^2 + v_i(S')} \\
&= \frac{\min_{j \in \{1,2\}} v_i(S_j)}{v_i^2 + v_i(S')} \\
&\leq \frac{\min_{j \in \{1,2\}} v_i(S_j) - v_i(S')}{v_i^2} \\
&\leq \frac{\min_{j \in \{1,2\}} v_i(S_j \setminus S')}{v_i^2} \\
&\leq \frac{\min_{j \in \{1,2\}}\{|S_j \setminus S'| \cdot \max\{v_i(\ell) : \ell \in S_j \setminus S'\}\}}{v_i^2}. \\
&\leq \frac{(m - k) \cdot \min_{j \in \{1,2\}} \max\{v_i(\ell) : \ell \in S_j\}}{v_i^2} \\
&\leq m - k.
\end{aligned}
$$

where the before last inequality is since if $S_j \subseteq S'$ for some $j$, then $v_i^2 + v_i(S') \geq \mu_i$; therefore $S_1 \setminus S'$ and $S_2 \setminus S'$ are two disjoint non empty subsets and $|S_1 \setminus S'| + |S_2 \setminus S'| = m - k + 1$, hence the maximum number of elements in one of these subsets is $m - k$. □

## D  Deferred proofs from Section 4

*Proof of Lemma 4.1.* By Observation 4.1, we have $v_i(a_i^k) \geq v_i^{2k}$, therefore

$$v_i(A_i) = \sum_{k=1}^{|A_i|} a_i^k \geq \sum_{k=1}^{\lfloor m/2 \rfloor} v_i^{2k}. \tag{1}$$

Since $i$'s favorite item must be absent from some set of the sets defining the MMS value,

$$\sum_{k=2}^{m} v_i^k \geq \mu_i.$$

Since the $v_i^k$ are ordered, $v_i^{2k} \geq v_i^{2k+1}$, hence $\sum_{k=1}^{\lfloor m/2 \rfloor} v_i^{2k} \geq \sum_{k=1}^{\lfloor m/2 \rfloor} v_i^{2k+1}$. Therefore,

$$\sum_{k=1}^{\lfloor m/2 \rfloor} v_i^{2k} \geq \mu_i/2 \qquad (2)$$

By Equations (1),(2), we have:

$$v_i(A_i) \geq \sum_{k=1}^{\lfloor m/2 \rfloor} v_i^{2k} \geq \mu_i/2.$$

$\square$

*Proof of Lemma 4.2.* Let $j_1$ be the first item assigned in `Balanced-Round-Robin` to agent 1. By definition, $j_1$ is agent 1's favorite item in $M$ according to $p_1$. Clearly, in `Plant-and-Steal`, $j_1$ is also agent 1's favorite item in $A_1 \subseteq M$ according to $p_1$. Hence, $\hat{j}_1 = j_1$. By the definition of `Plant-and-Steal`, $j_1 \in T_2$. Since we assume the prediction corresponds to agent 1's actual value, $j_1$ is also agent 1's favorite item in $T_2 \subseteq M$, which implies $\tilde{j}_1 = j_1$.

Similarly Let $j_2$ be the first item assigned in `Balanced-Round-Robin` to agent 2. By definition, $j_2$ is agent 2's favorite item in $M \setminus \{j_1\}$ according to $p_2$. Since $j_1 \in A_1$, $A_2 \subseteq M \setminus \{j_1\}$. Therefore, $j_2$ is also agent 2's favorite item in $A_2$ according to $p_2$. Hence, $\hat{j}_2 = j_2$. Since we established that $\hat{j}_1 = j_1$, we have that $T_1 \subseteq M \setminus \{j_1\}$ and $j_2 \in T_1$. Since we assume the prediction corresponds to agent 1's actual value, $j_2$ is also agent 1's favorite item in $T_1$, implying $\tilde{j}_2 = j_2$. We get that $X_1 = A_1$ and $X_2 = A_2$ as required. $\square$

## E  `1-2-RR-Plant-and-Steal` Mechanism: A $3/2$-consistent, $\lfloor 2m/3 \rfloor$-robust Mechanism

In this section, we show that using a modified round-robin allocation procedure to instantiate the `Plant-and-Steal` framework give an improved $3/2$ consistency guarantee, while maintaining $O(m)$ robustness. Consider the `Balanced-Round-Robin`allocation procedure depicted in Algorithm 2. One can show that the agent that picks first actually gets a value at least as large as their MMS, while for the second agent this analysis is indeed tight.[7] In order to compensate agent 2, `1-2-Round-Robin` lets this agent pick *two items* each round. See Algorithm 3 for details.

---
**ALGORITHM 3:** `1-2-Round-Robin`

**Input**  : Preference orders of agents over items $\mathbf{v} = (v_1, v_2)$.
**Output** : An allocation $A_1 \bigcup A_2 = M$.
$A_i \leftarrow \emptyset$, for every agent $i \in N$
**for** $r = 1, \ldots, \lceil |M|/3 \rceil$: **do**
 $\quad A_1 \leftarrow A_1 + v_1^*(M \setminus A_1 \setminus A_2)$
 $\quad A_2 \leftarrow A_2 + v_2^*(M \setminus A_1 \setminus A_2)$
 $\quad A_2 \leftarrow A_2 + v_2^*(M \setminus A_1 \setminus A_2)$

---

Let $a_i^k$ be agent $i$'s $k$th choice in `1-2-Round-Robin`, we observe the following.

**Observation E.1.** *The output $(A_1, A_2)$ of the 1-2-Round-Robin procedure, satisfies:*

1. *$|A_1| = \lceil \frac{m}{3} \rceil$ and $|A_2| = \lfloor \frac{2m}{3} \rfloor$.*
2. *$a_1^k \in \{v_1^\ell\}_{\ell \in [3k-2]}$, $a_2^{2k-1} \in \{v_2^\ell\}_{\ell \in [3k-1]}$ and $a_2^{2k} \in \{v_2^\ell\}_{\ell \in [3k]}$.*

---
[7]Consider the case where the agents' valuations are $(m-1, 1, \ldots, 1)$. According to Round-Robing allocation, the first item will be assigned to agent 1, and agent 2 will have $m/2$ items of value 1, while $\mu_2 = m - 1$.

Amanatidis et al. [6] show that `1-2-Round-Robin` guarantees each agent $2/3$ of their MMS. We provide the proof for completeness.

**Lemma E.1** (Amanatidis et al. [6]). *Let $(A_1, A_2)$ be the allocation of `1-2-Round-Robin`. For every agent $i$, $v_i(A_i) \geq 2\mu_i/3$.*

*Proof.* We first prove the approximation for player 1 (the first player to be allocated). First, observe that $v_1(M) \geq 2\mu_1$. Let $I_1 = \{v_1^{3k-2} : k = 1, \ldots, \lceil m/3 \rceil\}$ be the worst possible allocation agent 1 might get in the `1-2-Round-Robin` allocation. Notice that $v_1(I_1) \geq v_1(M)/3 \geq 2\mu_1/3$. By Observation E.1, $v_1(a_1^k) \geq v_1^{3k-2}$. Therefore, $v_1(A_1) \geq v_1(I_1) \geq 2\mu_1/3$.

Now consider player 2. As stated in the proof of Lemma 4.1, $v_2(M \setminus v_2^1) \geq \mu_2$. Let

$$I_2^a = \{v_2^{3k-1} : k \in \mathbb{N}_{>0} \ \wedge \ 3k-1 \leq m\} \text{ and } I_2^b = \{v_2^{3k} : k \in \mathbb{N}_{>0} \ \wedge \ 3k \leq m\}.$$

First, notice that

$$v_2(I_2^a \cup I_2^b) \ \geq \ 2v_2(M \setminus v_2^1)/3 \ \geq \ 2\mu_2/3.$$

Moreover, by Observation E.1, we have, $v_2(a_2^{2k-1}) \ \geq \ v_2^{3k-1}$, and $v_2(a_2^{2k}) \ \geq \ v_2^{3k}$. Therefore, $v_2(A_2) \geq v_2(I_2^a \cup I_2^b) \geq 2\mu_2/3$. $\qquad\square$

The mechanism we consider in this section, `1-2-RR-Plant-and-Steal`, results from instantiating `Plant-and-Steal` with `1-2-Round-Robin` as $\mathcal{A}$.

The next lemma states that if the predictions correspond to the preference orders of the real valuations, then `1-2-RR-Plant-and-Steal` outputs the same allocation as `1-2-Round-Robin`. The proof is omitted as it is identical to the proof of 4.2.

**Lemma E.2.** *When predictions correspond to actual values, `1-2-RR-Plant-and-Steal` outputs the same allocation as `1-2-Round-Robin`.*

We next show that in `1-2-RR-Plant-and-Steal` we are able to achieve a better consistency, while slightly weakening the robustness guarantee.

**Theorem E.1.** *Mechanism `1-2-RR-Plant-and-Steal` is truthful, $3/2$-consistent and $\lfloor \frac{2m}{3} \rfloor$-robust.*

*Proof.* By Lemma 3.1, the mechanism is truthful. By Observation E.1, each agent receives at least $\lceil m/3 \rceil$ items; combining with Lemma 3.4, we get that the mechanism is $\lfloor \frac{2m}{3} \rfloor$-robust. Finally, if predictions correspond to valuations, by Lemma E.1 and Lemma E.2, the allocation is $3/2$-approximation to the MMS. Thus, the mechanism is $2/3$-consistent. $\qquad\square$

# F   Deferred proofs from Section 5

*Proof of Lemma 5.1.* Let $S = \{s_1, \ldots, s_k\}$ ($|S| = k$) and $T = \{t_1, \ldots, t_\ell\}$ ($|T| = \ell$). We have the following.

$$
\begin{aligned}
v_i(S) \ &= \ \sum_{j=1}^{k} v_i(s_j) = \sum_{j=1}^{k} \int_0^\infty h_\tau(v_i(s_j)) d\tau = \int_0^\infty \sum_{j=1}^{k} h_\tau(v_i(s_j)) d\tau = \int_0^\infty v_i^\tau(S) d\tau \\
&\geq \ \int_0^\infty v_i^\tau(T)/c \, d\tau = \frac{1}{c} \int_0^\infty \sum_{j=1}^{\ell} h_\tau(t_j) d\tau = \frac{1}{c} \sum_{j=1}^{\ell} \int_0^\infty h_\tau(t_j) d\tau = \frac{1}{c} \sum_{j=1}^{\ell} v_i(t_j) \\
&= \ v_i(T)/c,
\end{aligned}
$$

where we use the identity $\int_0^\infty h_\tau(q) d\tau = q$.

$\qquad\square$

*Proof of Lemma 5.2.* Let $\lfloor \frac{m}{2} \rfloor \leq m_i \leq \lceil \frac{m}{2} \rceil$ be the number of items agent $i$ gets by Mechanism `B-RR-Plant-and-Steal`. Let $A_i = \{a_i^1, a_i^2, \ldots, a_i^{m_i}\}$ be the items assigned to agent $i$ in the

Round-Robin according to the predicted orderings $\mathbf{p}$, where $a_i^\ell$ is the item allocated to $i$ in the $\ell^{\text{th}}$ round of Round-Robin. First, by Observation 4.1, we have:

$$a_i^\ell \in \{p_i^j\}_{j \in \{1,\ldots,2\ell\}}. \tag{3}$$

For a fixed $\tau \geq 0$, let $L_\tau = v_i^\tau(R_i)$ be the number of values larger than threshold $\tau$ in $R_i$. We show that if the Kendall tau distance is at most $d$, then it must be the case that

$$v_i^\tau(A_i) \geq L_\tau - \sqrt{d}. \tag{4}$$

Note that $h_\tau(v_i^k) = 1$ for $k \leq 2 \cdot L_\tau$ since $R_i$ gets every second item by the sorted values of agent $i$. This implies that if $h_\tau(v_1(a_i^\ell)) = 0$ then $a_i^\ell = v_i^k$ for $k > 2 \cdot L_\tau$. Moreover, if $\ell \leq L_\tau$ then $a_i^\ell = p_i^k$ for $k \leq 2 \cdot L_\tau$ by Eq. (3). Thus, if $\sum_{k=1}^{L_\tau} h_\tau(v_1(a_i^k)) < L_\tau - \sqrt{d}$ there are *strictly* more than $\lceil \sqrt{d} \rceil$ items whose rank according to the true valuation is at most $2 \cdot L_\tau$, and their rank according to the prediction is at least $2 \cdot L_\tau + 1$. We show that this implies that the Kendall tau distance is larger than $d$, yielding a contradiction. Formally, let

$$G_1 = \{v_1^k\}_{k \in \{1,\ldots,2 \cdot L_\tau\}} \setminus \{p_1^k\}_{k \in \{1,\ldots,2 \cdot L_\tau\}}$$

be the set of items whose rank is at most $2 \cdot L_\tau$ according to the real values but not according to the predictions, and let

$$G_2 = \{v_1^k\}_{k \in \{2 \cdot L_\tau + 1,\ldots,m\}} \setminus \{p_1^k\}_{k \in \{2 \cdot L_\tau + 1,\ldots,m\}}$$

be the set of items whose rank is strictly larger than $2 \cdot L_\tau$ according to the real values but not according to the predictions. By the above, $|G_1| = |G_2| > \lceil \sqrt{d} \rceil$, and for each pair $j \in G_1, j' \in G_2$,

1. $j$ rank according to $v_i$ is at most $2 \cdot L_\tau$ and $j'$ rank according to $v_i$ is at least $2 \cdot L_\tau + 1$;

2. $j'$ rank according to $p_i$ is at most $2 \cdot L_\tau$ and $j$ rank according to $p_i$ is at least $2 \cdot L_\tau + 1$.

That is, $j$ and $j'$ are ordered oppositely in the ordering according to $p_i$ and $v_i$. Since there are $|G_1| \cdot |G_2| > d$ such pairs, we get that the Kendall tau distance is *strictly* greater than $d$, a contradiction. $\qquad\square$

*Proof of Theorem 5.1.* Recall that, in Eq. (2) of Lemma 4.1, we establish that:

$$v_i(R_i) \geq \mu_i/2 \tag{5}$$

In addition, we apply the zero-one principle to show that Lemma 5.1 holds for the sets $X_i$ and $R_i$ with $c = \sqrt{d} + 3$. The proof follows by combining these two results.

Notice that $|A_i \setminus X_i| \leq 2$, because in the "stealing" phase, agent $i$ might not take the "planted" item from $A_i$ back, and the other agent might take one item from $A_i$.[8] Moreover, by Lemma 3.2, either $v_i^1$ or $v_i^2$ are in $X_i$. Therefore, for every threshold $\tau \geq 0$,

$$
\begin{aligned}
v_i^\tau(X_i) &\geq \max\{h_\tau(v_i^2), v_i^\tau(A_i) - 2\} \\
&\geq \max\{h_\tau(v_i^2), v_i^\tau(R_i) - \sqrt{d} - 2\}, \tag{6}
\end{aligned}
$$

where the inequality follows Lemma 5.2.

If $h_\tau(v_i^2) = 0$, then $v_i^\tau(R_i) \leq |R_i| \cdot h_\tau(v_i^2) = 0$, and Lemma 5.1 holds with $c = 0$. Therefore, the interesting case is when $h_\tau(v_i^2) = 1$. Consider the ratio $\frac{v_i^\tau(R_i)}{v_i^\tau(X_i)}$ which we want to bound. Since $v_i^\tau(X_i) \geq h_\tau(v_i^2) = 1$, $v_i^\tau(R_i) \in [1, \sqrt{d} + 3]$ implies that

$$\frac{v_i^\tau(R_i)}{v_i^\tau(X_i)} \leq v_i^\tau(R_i) \leq \sqrt{d} + 3.$$

On the other hand, by Eq. (6), setting $v_i^\tau(R_i) = \sqrt{d} + 3 + \delta$ for $\delta > 0$ implies that $v_i^\tau(X_i) \geq v_i^\tau(R_i) - \sqrt{d} - 2 \geq 1 + \delta$, which yields

$$\frac{v_i^\tau(R_i)}{v_i^\tau(X_i)} \leq \frac{\sqrt{d} + 3 + \delta}{1 + \delta} \leq \sqrt{d} + 3.$$

We get that Lemma 5.1 holds for $X_i$ and $R_i$ with $c = \sqrt{d} + 3$. Thus,

$$v_i(X_i) \geq v_i(R_i)/(\sqrt{d} + 3) \geq \mu_i/(2\sqrt{d} + 6),$$

where the last inequality follows Eq. (5). $\qquad\square$

---

[8] In fact, this holds for any noise in the valuations of the other agent.

# G  Non-ordering Predictions

In this Section, we consider the case where predictions are not necessarily preference orders over items. In Section G.1, we show that for any prediction the mechanism might get, consistency is bounded away from 1. Sections G.2, G.3, we study *succinct* predictions, i.e. predictions about general structure of the preferences of two agents. Section G.2 presents a 4-consistent and $\lceil m/2 \rceil$-robust mechanism, whose consistency relies on the correctness of only a $\log m$-bit prediction about the preferences of the two agents. In Section G.3, we show that a $2 + \epsilon$-consistent and $\lceil m/2 \rceil$-robust mechanism exists, whose consistency relies on correctly predicting only $O(\log m/\epsilon)$ bit about the preferences of the two agents.

## G.1  No Mechanism with $< 6/5$ Consistency and Bounded Robustness

In [7] they define the following family of mechanisms.

**Definition G.1** (Singleton Picking-Exchange Mechanisms [7]). *A mechanism $X$ is a singleton picking-exchange mechanism if for each $i \in \{1, 2\}$, there is exactly one of two sets: either $N_i \subseteq M$, or $E_i = \{\ell_i\}$ for a single item $\ell_i \in M$. If $N_i$ is non-empty, then the mechanism lets player $j \neq i$ pick item $\ell \in N_i$ that maximizes $v_j(\ell)$, and $i$ gets $N_i \setminus \{\ell\}$. If both $E_1, E_2$ are non-empty, then the agents exchange the two items $\ell_1 \in E_1$ and $\ell_2 \in E_2$ if $v_1(\ell_2) > v_1(\ell_1)$ and $v_2(\ell_1) > v_1(\ell_2)$. Notice that if $m > 2$, either $E_1$ or $E_2$ is empty and there will be no exchange.*

[7] showed the following.

**Lemma G.1.** *In order for a mechanism to be truthful and have a bounded approximation, it has to be a singleton picking-exchange mechanism*

We make use of this characterization in our impossibility.

**Theorem G.1.** *For any $\epsilon > 0$, there is no truthful a mechanism with consistency $6/5 - \epsilon$ and bounded robustness.*

*Proof.* Consider the case where $p_1 = p_2 = (1/2, 1/2, 1/3, 1/3, 1/3)$. Notice that for the predictions, $\mu_1 = \mu_2 = 1$. We show that for any singleton-picking-exchange mechanism, no agent obtains both large items (of value $1/2$). Consider agent 1 (the argument is symmetric for agent 2). If $N_1$ is non-empty, then if both large items are in $N_1$, surely 1 will only get one of them. If both large items are in $N_2$, then agent 2 will surely pick one of them, and agent 1 will only get one of them. If one large item is in $N_1$ and the other is in $N_2$, each agent $i$ will pick the large item in $N_i$. If agent 2 has a large item in $E_2$, then since $N_1$ is non-empty, $E_1$ is empty and agent 2 will keep the large item. Now consider the case where $E_1$ is non-empty. In this case, $E_1$ contains one item, and $N_1$ is empty. Since $E_2$ can contain at most one item, and there are more than 2 items, in this case, $E_2 = \emptyset$, and $|N_2| = 4$. Therefore, $N_2$ contains at least one large item. Since agent 2 will always pick the large item, agent one only gets one large item. We conclude that for any singleton picking-exchange mechanism, the large items are split among the agents. Since there are 3 small items, there must be an agent that gets at most one small item, and this agent has an overall value of at most $1/2 + 1/3 = 5/6$, while the MMS is 1. Thus the claim follows. $\qquad\square$

## G.2  $4$-Consistent, $(m-1)$-Robust Mechanism Using a $3 \log m + 1$-Space Prediction

Let us formally define a mechanism that uses a space-$s$ prediction

**Definition G.2.** *A learning-augmented mechanism is a space-$s$ mechanism if the prediction space $\mathcal{P}$ can be represented by the elements of $\{0, 1\}^s$.*

We first give a simple mechanism that only requires $3 \log m + 1$ bits of information about the valuations $v_1$ and $v_2$. It will only need to know an index $j_0$ in $[m]$ together with a bit $b$ to produce an approximately-optimal allocation, and an additional $2 \log n$ bits to implement the planting phase. The mechanism will utilize the `Plant-and-Steal` framework in conjunction with the well-known bag-filling allocation procedure:

We see that, in order to predict the behaviour of the mechanism above, one only needs to predict accurately the index $j_0$ on which the mechanism terminates, as well as a bit $b \in \{1, 2\}$ that encodes

---

**MECHANISM 4:** `Bag-Filling`

---

**Input** : Preference orders of agents over items $\mathbf{v} = (v_1, v_2)$ on a set of items $M = [m]$
**Output** : Allocations $A_1 \bigcup A_2 = M$
$j \leftarrow 1$
**for** $j = 1, \ldots, m$: **do**
    **if** $\frac{v_1([j])}{v_1([m])} \geq \frac{1}{2}$ **then** Output $(A_1, A_2) \leftarrow ([j], [m] \setminus [j])$ and terminate
    **if** $\frac{v_2([j])}{v_2([m])} \geq \frac{1}{2}$ **then** Output $(A_1, A_2) \leftarrow ([m] \setminus [j], [j])$ and terminate

---

whether the algorithm terminates due to the condition $\frac{v_1([j])}{v_1([m])} \geq \frac{1}{2}$ being satisfied or due to the condition $\frac{v_2([j])}{v_2([m])} \geq \frac{1}{2}$ being satisfied. This can be encoded using $\log m + 1$ bits.

We also see that the The `Plant-and-Steal` framework when used with the `Bag-Filling` allocation procedure gives a truthful 4-consistent and a $m - 1$-robust[9] allocation mechanism. The truthfulness and robustness follow immediately from Lemmas 3.1 and 3.4 respectively.

The 4-consistency holds for the following reason. It is a well-known fact (see i.e. [10]) that the partition $(A_1, A_2)$ given by the bag-filling algorithm satisfies $v_1(A_1) \geq \mu_1/2$ and $v_2(A_2) \geq \mu_2/2$. By inspecting the `Plant-and-Steal` framework (Algorithm 1), we see that both agent 1 and agent 2 will either (i) retain their most preferred item in $A_1$ and $A_2$ respectively or (ii) Lose this item, but obtain an item that they prefer even more. Overall, this implies that in the worst case the difference $v_1(A_1) - v_1(X_1)$ will equal to the value of the second-most favorite item of Agent 1 in $A_1$. This implies that $v_1(X_1) \geq \frac{1}{2} v_1(A_1) \geq \frac{\mu_1}{4}$. Analogously, we see that $v_1(X_2) \geq \frac{1}{2} v_1(A_2) \geq \frac{\mu_2}{4}$.

## G.3   $2 + \epsilon$-Consistent, $\lceil \frac{m}{2} \rceil$-Robust Mechanism Using a $O(\log m/\epsilon)$-Space Prediction

We now show that a better consistency of $2 + \epsilon$ can be achieved at the cost predicting $O(\log m/\epsilon)$ bits of information about the valuations $v_1$ and $v_2$. We will also obtain a better robustness of $\lceil \frac{m}{2} \rceil$. To do this, we will use the `Plant-and-Steal` framework in conjunction with the `Cut-and-Balance` allocation procedure. We first explain how the mechanisms above can be implemented by only

---

**ALGORITHM 5:** `Cut-and-Balance`

---

**Output** : Allocations $A_1 \bigcup A_2 = M$
Consider a partition $S_1 \bigcup S_2 = M$ satisfying $|S_1| \geq |S_2|$ and

$$\min_{j \in \{1,2\}} v_1(S_j) \geq (1 - \epsilon) \max_{T_1 \bigcup T_2 = M} \min_{j \in \{1,2\}} v_1(T_j) = (1 - \epsilon)\mu_1$$

Let $S' \subset S_1$ be a set of $\lfloor m/2 \rfloor - |S_2|$ items satisfying
   • $v_1(S') \leq v_1(S_1)/2$
   • if $|S_2| > 1$ additionally satisfying $v_1(S') \leq v_1(S_1 \setminus \{\hat{j}, \hat{j}'\})/2$, for some
     $\hat{j} \in \arg\max_{j \in S_1} v_1(j)$ and $\hat{j}' \in \arg\max_{\ell \in S_1 \setminus \hat{j}} v_1(\hat{j})$

Set $\tilde{S}_1 \leftarrow S_1 \setminus S'$ and $\tilde{S}_2 \leftarrow S_2 \cup S'$
Let $i_2 \leftarrow \arg\max_{i \in \{1,2\}} p_2(\tilde{S}_i)$ and let $i_1$ be the index of the other bundle
Set $A_1 \leftarrow S_{i_1}$ and $A_2 \leftarrow S_{i_2}$, and output the allocation $(A_1, A_2)$

---

obtaining $O(\log m/\epsilon)$ bits of information about the valuations $v_1$ and $v_2$. This follows from the following proposition, the proof of which is given in Appendix G.4.

**Proposition G.1.** *Suppose $M = [m]$. There is a partition $M = L_1 \bigcup L_2 \bigcup S$ and indices $\alpha_1, \beta_1, \alpha_2$ and $\beta_2$ with $|L_1| + |L_2| \leq O\left(\frac{1}{\epsilon}\right)$, such that the partition $M = S_1 \bigcup S_2$ defined as $S_1 = L_1 \bigcup (S \bigcap [\alpha_1, \beta_1])$ and $S_2 = L_2 \bigcup (S \bigcap [\alpha_2, \beta_2])$ satisfies $|S_1| \geq |S_2|$ and $\min(v_1(S_1), v_1(S_2)) \geq (1 - \epsilon/4)\mu_1$.*

*Additionally, there exist integers $\alpha_3, \beta_3, \alpha_4$ and $\beta_4$ such that the set $S' = S \bigcap ([\alpha_3, \beta_3] \bigcup [\alpha_4, \beta_4])$ satisfies $|S'| = \lfloor m/2 \rfloor - |S_2|$, $S' \subset S_1$, $v_1(S') \leq v_1(S_1)/2$ and if $|S_2| > 1$ then $S'$ also satisfies $v_1(S') \leq v_1(S_1 \setminus \{\hat{j}, \hat{j}'\})/2$, where $\hat{j} \in \arg\max_{j \in S_1} v_1(j)$ and $\hat{j}' \in \arg\max_{j \in S_1 \setminus \hat{j}} v_1(j)$.*

---

[9]Note that $\min(|A_1|, |A_2|) \leq m - 1$ which implies that the algorithm is $(m - 1)$-robust.

The main ideas for proving Proposition G.1 are: (i) using the sets $L_1$ and $L_2$ to handle elements $x$ whose value $v(x)$ is large, and separate the remaining items into the set $S$ (ii) Showing that the remaining items can be separated into well-behaved subsets of the form $S \bigcap [\alpha_i, \beta_i]$.

The proposition above implies that the sets $S_1, S_2$ and $S'$ can be represented exactly via sets $L_1$ and $L_2$, together with the indices $\{\alpha_1, \cdots, \alpha_4, \beta_1, \cdots \beta_4\}$. We will also need to know the index $i_2 \in \{1, 2\}$. Since the sets $L_1$ and $L_2$ have a size of $O(1/\epsilon)$, all this information amounts to $O(\log m/\epsilon)$ bits as claimed.

The following proposition implies the truthfulness, the robustness and the consistency of the mechanism that combines the `Cut-and-Balance` allocation procedure with the `Plant-and-Steal` framework.

**Theorem G.2.** *The `Plant-and-Steal` framework, when used with `Cut-and-Balance` allocation procedure, gives a truthful, $2 + \epsilon$-consistent and a $\lceil m/2 \rceil$-robust allocation mechanism.*

*Proof.* Truthfulness follows from Lemma 3.1. Since the sets $A_1$ and $A_2$ both have size at most $\lceil m/2 \rceil$, the robustness follows via Lemma 3.4.

The proof of $(2 + \epsilon)$-consistency is deferred to Appendix G.5. The main challenge for showing the bound on consistency is the fact that both the `Cut-and-Balance` allocation procedure and the `Plant-and-Steal` framework can reduce the consistency by a factor of 2. Naively, one would expect the overall consistency to be close to 4, given that each stage can lose a factor of 2 in consistency. However, our insight is that for the instances, on which the `Cut-and-Balance` allocation procedure loses a factor of 2 in consistency, the `Plant-and-Steal` framework will have consistency close to 1, and vice versa. This allows us to prove a tighter bound of $2 + \epsilon$ on the consistency of our overall algorithm. □

## G.4 Proof of Proposition G.1

We first show the following, which implies the first half of Proposition G.1.

**Proposition G.2.** *There exists a partition $M = L_1 \bigcup L_2 \bigcup S$ and and indices $\alpha_1, \alpha_2, \beta_1, \beta_2$ in $[m]$ such that $M = [\alpha_1, \beta_1] \bigcup [\alpha_2, \beta_2]$, for the sets $S_1 = L_1 \cup (S \cap [\alpha_1, \beta_1])$ and $S_2 = L_2 \cup (S \cap [\alpha_2, \beta_2])$ we have*

- $\min\{v_1(S_1), v_1(S_2)\} \geq (1 - \epsilon/8)\mu_1$

- $|L_1| + |L_2| \leq \lceil \frac{8}{\epsilon} \rceil + 2$

- $|S_1| \geq |S_2|$.

- *For every $x$ in $L_1$ and $y$ in $S_1$ we have $v_1(x) > v_1(y)$. Analogously, for every $x$ in $L_2$ and $y$ in $S_2$ we have $v_1(x) > v_1(y)$*

- *There are $\hat{j}, \hat{j}' \in L_1$ satisfying $\hat{j} \in \arg\max_{\ell \in S_1} v_1(\ell)$ and $\hat{j}' \in \arg\max_{\ell \in S_1 \setminus \hat{j}} v_1(\ell)$,*

We do this by inspecting two types of items, large items, with value greater than $\epsilon\mu_1/4$, and small items items with value at most $\epsilon\mu_1/4$. We first show that there are $O(1/\epsilon)$ large items, therefore, separating these items into two bundles require at most $O(1/\epsilon)$ intervals. Moreover, we can find a separation of the larges items into two sets, $L_1, L_2$, and a single index $j \in [m]$ such that all small items to the left of $j$ (including) together with $L_1$ form $S_1$, and all items to the right of $j$ (excluding) together with $L_2$ form $S_2$, such that $S_1, S_2$ satisfy the approximation requirement. It is easy to see that this increases the number of intervals by at most 1.

We start by showing there are not too many large items.

**Lemma G.2.** *There are at most $\lceil \frac{8}{\epsilon} \rceil$ items with value strictly greater than $\epsilon\mu_1$ for agent 1.*

*Proof.* Let items with value greater than $\epsilon\mu_1/4$ be the *large* items. Suppose there are at least $\lceil \frac{8}{\epsilon} \rceil + 1$ large items. If $\lceil \frac{8}{\epsilon} \rceil$ is even, consider a partition $(S_1, S_2)$ such that each $S_i$ gets at least $\lceil \frac{8}{\epsilon} \rceil/2$ large items and the rest are allocated arbitrarily. If $\lceil \frac{8}{\epsilon} \rceil$ is odd, consider the allocation in which each $S_i$ gets $(\lceil \frac{8}{\epsilon} \rceil + 1)/2$ large items and the rest are allocated arbitrarily. In either case, each $S_i$ gets at least $\lceil \frac{8}{\epsilon} \rceil/2 \geq \frac{4}{\epsilon}$ large items. Thus, $\min\{v_1(S_1), v_1(S_2)\} > \epsilon\mu_1/4 \cdot \frac{4}{\epsilon} = \mu_1$, a contradiction. □

We are now ready to prove Proposition G.2.

*Proof of Proposition G.2.* Consider the set of large items, $L = \{j \in [m] \; : \; v_1(j) > \epsilon\mu_1/4\}$, and let $S = M \setminus L$ be the set of small items.

We give a constructive proof which finds both sets $L_1, L_2$ and an index $j$ satisfying the condition stated in the lemma. Let

$$(L_1, L_2) \in \arg\max_{(T_1, T_2) \; : \; T_1 \bigcup T_2 = L} \min_{j \in \{1,2\}} v_1(S_j).$$

We use the following procedure to find $j$.

1. Let $j_\ell = 0$ and $j_r = m$.

2. While $j_\ell \neq j_r$:

    (a) Let $S_\ell = L_1 \cup \{j' \in S \; : \; j' \leq j_\ell\}$ and $S_r = L_2 \cup \{j' \in S \; : \; j' > j_r\}$.
    (b) If $v_1(S_\ell) < v_1(S_r)$ :
        - $j_\ell := j_\ell + 1$.
    (c) Else:
        - $j_r := j_r - 1$.

3. Set $j := j_\ell = j_r$.

We consider two cases:

**Case 1:** $j = 0$ (or symmetrically, $j = m$). Without loss of generality, suppose that $j = m$. We first show that if $v_1(S_1) < v_1(S_2)$ then $\min\{v_1(S_1), v_1(S_2)\} = \mu_1$. Notice that since $S_1$ gets all the small items, it must be the case that $v_1(L_1) < v_1(L_2)$. Suppose there's a different partition $T_1 \bigcup T_2$ such that $\min\{v_1(T_1), v_1(T_2)\} > \min\{v_1(S_1), v_1(S_2)\}$. Without loss of generality, let $v_1(T_1 \cap L) \leq v_1(T_2 \cap L)$ (otherwise, we can rename both bundles). By the definition of $L_1, L_2$, it must be the case that $v_1(L_1) \geq v_1(T_1 \cap L)$. Thus, Since $T_1 \setminus (T_1 \cap L) \subseteq S$, it must be that

$$v_1(S_1) \; = \; v_1(L_1) + v_1(S) \; \geq \; v_1(T_1 \cap L) + v_1(T_1 \setminus (T_1 \cap L)) \; = \; v_1(T_1) \; \geq \; \min\{v_1(T_1), v_1(T_2)\},$$

a contradiction.

On the other hand, if $v_1(S_1) \geq v_1(S_2) = v_1(L_2)$, by condition 2b of the above procedure, it must be the case that when $j_\ell$ was equal $m - 1$,

$$v_1(S_\ell) \; < \; v_1(S_r) \; = \; v_1(L_2) \; = \; v_1(S_2).$$

Thus,

$$v_1(S_1) \; = \; v_1(S_\ell) + v_1(m) \; < \; v_1(S_2) + \epsilon\mu_1/4.$$

We get that

$$v_1(S_2) \; \geq \; v_1(S_1) - \epsilon\mu_1/4 \; \geq \; 2\mu_1 - v_1(S_2) - \epsilon\mu_1/4 \; \Rightarrow$$
$$\min\{v_1(S_1), v_1(S_2)\} \; = \; v_1(S_2) \; \geq \; (1 - \epsilon/8)\mu_1, \tag{7}$$

where the second inequality follows since $2\mu_1 \leq v_1(S_1) + v_1(S_2)$.

Case 2: $0 < j < m$. In this case, since both $j_\ell$ and $j_r$ were moved, there were some values of $j_\ell$ and $j_r$ such that $v_1(S_\ell) \leq v_1(S_r)$ and some values such that $v_1(S_\ell) > v_1(S_r)$. Assume initially that $v_1(S_\ell) \leq v_1(S_r)$. Since at each step of the procedure, the the lower-valued bundle can increase by at most $\epsilon\mu_1/4$, when the first item is added to $S_\ell$ such that $v_1(S_\ell) > v_1(S_r)$, it must be the case that $v_1(S_\ell) \leq v_(S_r) + \epsilon\mu_1/4$. It is easy to see that the invariant where $|v_1(S_\ell) - v_1(S_r)| \leq \epsilon\mu_1/4$ is kept throughout the run of the procedure. Therefore, this also holds for the final $S_1$ and $S_2$. Thus, we can use the same reasoning of Eq. (7) to conclude that $\min\{v_1(S_1), v_1(S_2)\} \geq (1 - \epsilon/8)\mu_1$.

Thus, the sets $S_1$ and $S_2$ have a form $S_1 = L_1 \cup (S \cap [1, j])$ and $S_2 = L_2 \cup (S \cap [j+1, m])$ and have the form required. If $|S_1| < |S_2|$ we can swap our definitions for the sets $S_1$ and $S_2$, thus ensuring that $|S_1| > |S_2|$. Due to our definitions of $L_1$ and $L_2$ we have for every $x$ in $L_1$ and $y$ in $S_1$ we have $v_1(x) > v_1(y)$. Analogously, for every $x$ in $L_2$ and $y$ in $S_2$ we have $v_1(x) > v_1(y)$.

We can ensure that There are $\hat{j}, \hat{j}' \in L_1$ satisfying

$$\hat{j} \in \arg\max_{\ell \in S_1} v_1(\ell) \text{ and } \hat{j}' \in \arg\max_{\ell \in S_1 \setminus \hat{j}} v_1(\ell),$$

by adding such values from $S \cap [\alpha_1, \beta_1]$ to $L_1$ (we see that after this all other properties still hold). Overall, we see that $|L_1| + |L_2| \leq \lceil \frac{8}{\epsilon} \rceil + 2$, as required. $\square$

Now, we proceed to proving the second half of Proposition G.1. We will need the following lemma.

**Lemma G.3.** *Let $k_1$ and $k_2$ be positive integers satisfying $k_1 > k_2$, and let $f$ be a function mapping $[k_1]$ to non-negative real numbers. Then, there exist a pair of integers $\alpha, \beta, \alpha'$ and $\beta'$ in $[k_1]$ such that $|[\alpha, \beta] \cup [\alpha', \beta']| = k_2$ and*

$$\frac{\sum_{i \in [\alpha,\beta] \cup [\alpha',\beta']} f(i)}{k_2} \leq \frac{\sum_{i \in [k_1]} f(i)}{k_1}$$

*Proof.* We prove the lemma using the probabilistic method. Let $j$ be a uniformly random integer in $[k_1]$, and choose $\alpha, \beta, \alpha'$ and $\beta'$ such that

$$[\alpha, \beta] \cup [\alpha', \beta'] = \{j, j+1 \mod k_1, \cdots, j+k_2-1 \mod k_1\}.$$

We see that indeed a set chosen as above can be represented as a union of two intervals. Now, since $j$ is chosen uniformly at random form $[k_1]$, we see that for every element $i$ in $[k_1]$ we have

$$\Pr_{j \sim [k_1]}[i \in \{j, j+1 \mod k_1, \cdots, j+k_2-1 \mod k_1\}] = \frac{k_2}{k_1}.$$

Thus via linearity of expectation we have:

$$\mathbb{E}_{j \sim [k_1]}\left[\frac{1}{k_2} \sum_{i \in \{j, j+1 \mod k_1, \cdots, j+k_2-1 \mod k_1\}} f(i)\}\right] = \frac{1}{k_1} \sum_{i \in [k_1]} f(i).$$

Thus, since $f(i)$ is non-negative for all values of $i$, we see that for some specific choice of $j$ it has to be the case that

$$\frac{1}{k_2} \sum_{i \in \{j, j+1 \mod k_1, \cdots, j+k_2-1 \mod k_1\}} f(i)\} \leq \frac{1}{k_1} \sum_{i \in [k_1]} f(i),$$

which finishes the proof. $\square$

Now, we apply the lemma above. If $m < 4\lceil \frac{t}{\epsilon} \rceil + 2$ we can satisfy Proposition G.2 by:

1. First choosing a partition $M = S_1 \bigcup S_2$ such that $\min(v_1(S_1), v_1(S_2)) \geq \mu_1$ and $|R_1| \geq |R_2|$.

2. Define $L_2 := S_2$, put the smallest $\lfloor m/2 \rfloor - |S_2|$ elements of $S_1$ into $S$, and define $L_1$ to contain the rest of elements in $S_1$.

3. Define $\alpha_1 = \alpha_3 = 1$, $\beta_1 = \beta_3 = m$, $\alpha_2 = \beta_2 = \alpha_4 = \beta_4 = m+1$.

Overall, this allocates $S'$ to be the bottom $\lfloor m/2 \rfloor$ elements of $S_1$. We see that this suffices to guarantee the properties that $S'$ needs to satisfy in Proposition G.1.

Therefore, henceforth we can assume that $m > 4\lceil \frac{t}{\epsilon} \rceil + 2$. Since $|S_1| \geq m/2$ and $|L_1| \leq \frac{8}{\epsilon} + 2$, and $S_1 = L_1 \cup (S \cap [\alpha_1, \beta_1])$ this implies that $|S \cap [\alpha_1, \beta_1])| > 3\lceil \frac{t}{\epsilon} \rceil > m/2$ Thus, we can ensure that $|S'| = \lfloor m/2 \rfloor - |S_2|$ using a subset $S' \subset S \cap [\alpha_1, \beta_1])$.

If $|S_2| = 1$ we only need choose $S'$ to satisfy $|S'| = \lfloor m/2 \rfloor - |S_2|$ and $v_1(S') \leq v_1(S_1 \setminus \{\hat{j}, \hat{j}'\})/2$. First of all, since every element in $L_1$ is larger than any element in $S \cap [\alpha_1, \beta_1])$, we see that this is also true in average

$$\frac{\sum_{\ell \in S_1} v_1(\ell)}{|S_1|} \leq \frac{\sum_{\ell \in S \cap [\alpha_1, \beta_1])} v_1(\ell)}{|S \cap [\alpha_1, \beta_1])|} \tag{8}$$

Then, applying Lemma G.3 to the set $S \cap [\alpha_1, \beta_1])$ we see that there exist disjoint subsets $[\alpha_3, \beta_3]$ and $[\alpha_4, \beta_4]$ of $[\alpha_1, \beta_1]$ such that $|S \cap ([\alpha_3, \beta_3] \bigcup [\alpha_4, \beta_4]))| = \lfloor m/2 \rfloor - |S_2|$

$$\frac{\sum_{\ell \in S \cap [\alpha_1, \beta_1])} v_1(\ell)}{|S \cap [\alpha_1, \beta_1])|} \leq \frac{\sum_{\ell \in S \cap ([\alpha_3, \beta_3] \bigcup [\alpha_4, \beta_4]))} v_1(\ell)}{|S \cap ([\alpha_3, \beta_3] \bigcup [\alpha_4, \beta_4]))|} \tag{9}$$

Combing Equations 8 and 9, we see that taking $S' = S \cap ([\alpha_3, \beta_3] \bigcup [\alpha_4, \beta_4])$ satisfies $v_1(S') \leq v_1(S_1)/2$ and the other requirements of Proposition G.1.

Now, suppose $|S_2| > 1$. Since by Proposition G.2, the set $S$ does not contain the two largest elements $\hat{j}$ and $\hat{j}'$ of $S_1$, as well as the fact that every element in $L_1$ is at least as large as any element in $S \cap [\alpha_1, \beta_1])$, we see that every every element in $S_1 \setminus \{\hat{j}, \hat{j}'\}$ is either in $S \cap [\alpha_1, \beta_1])$ or greater than every element in $S \cap [\alpha_1, \beta_1])$. this implies that:

$$\frac{\sum_{\ell \in S_1 \setminus \{\hat{j}, \hat{j}'\}} v_1(\ell)}{|S_1| - 2} \leq \frac{\sum_{\ell \in S \cap [\alpha_1, \beta_1])} v_1(\ell)}{|S \cap [\alpha_1, \beta_1])|} \tag{10}$$

Then, we can again apply applying Lemma G.3 to the set $S \cap [\alpha_1, \beta_1])$ we see that there exist disjoint subsets $[\alpha_3, \beta_3]$ and $[\alpha_4, \beta_4]$ of $[\alpha_1, \beta_1]$ such that $|S \cap ([\alpha_3, \beta_3] \bigcup [\alpha_4, \beta_4]))| = \lfloor m/2 \rfloor - |S_2|$

$$\frac{\sum_{\ell \in S \cap [\alpha_1, \beta_1])} v_1(\ell)}{|S \cap [\alpha_1, \beta_1])|} \leq \frac{\sum_{\ell \in S \cap ([\alpha_3, \beta_3] \bigcup [\alpha_4, \beta_4]))} v_1(\ell)}{|S \cap ([\alpha_3, \beta_3] \bigcup [\alpha_4, \beta_4]))|} \tag{11}$$

Combing Equations 10 and 11, we see that taking $S' = S \cap ([\alpha_3, \beta_3] \bigcup [\alpha_4, \beta_4])$ satisfies $v_1(S') \leq v_1(S_1 \setminus \{\hat{j}, \hat{j}'\})/2$ as required in Proposition G.1. Note that this also implies that $v_1(S') \leq v_1(S_1\})/2$ since $\hat{j}, \hat{j}'$ have the top two largest values of $v_1$ in $S_1$. Overall, this finishes the proof of Proposition G.1.

### G.5 Proof of $(2 + \epsilon)$-consistency.

It remains to show that the algorithm is $2 + \epsilon$-consistent. We will be referencing the variables $\hat{j}_1, \hat{j}_2, \tilde{j}_1, \tilde{j}_2, T_1$ and $T_2$ within the Plant-And-Steal framework (Algorithm 1).

We first reason about agent 2. First, notice that since agent 2 has a higher value for $\tilde{S}_{i_2}$,

$$v_2(\tilde{S}_{i_2}) \geq \mu_2.$$

Since the mechanism had a chance to pick item $\hat{j}_2$ from $T_1$ as $\tilde{j}_2$, it must be the case that $v_2(\tilde{j}_2) \geq v_2(\hat{j}_2)$ (and possibly $\tilde{j}_2 = \hat{j}_2$). If $\tilde{j}_1 = \hat{j}_1$, then $T_2 \setminus \tilde{j}_1 = \tilde{S}_{i_2} \setminus \hat{j}_2$, and

$$\mu_2 \leq v_2(\tilde{S}_{i_2}) = v_2(\tilde{S}_{i_2} \setminus \hat{j}_2) + v_2(\hat{j}_2) \leq v_2(T_2 \setminus \tilde{j}_1) + v_2(\tilde{j}_2) = v_2(X_2).$$

Otherwise, $\tilde{j}_1 \in \tilde{S}_{i_2}$, and

$$\tilde{S}_{i_2} \setminus \hat{j}_2 \setminus \tilde{j}_1 \subset T_2 \setminus \tilde{j}_1 \ \Rightarrow\ v_2(\tilde{S}_{i_2} \setminus \hat{j}_2 \setminus \tilde{j}_1) \leq v_2(T_2 \setminus \tilde{j}_1). \tag{12}$$

Since $\hat{j}_2$ is the item with the highest value for agent 2 in $\tilde{S}_{i_2}$, $v_2(\tilde{j}_2) \geq v_2(\hat{j}_2) \geq v_2(k_1)$. Combining with Eq. (12), we get that

$$v_2(T_2 \setminus \tilde{j}_1 \cup \{\tilde{j}_2\}) \geq v_2(\tilde{S}_{i_2} \setminus \hat{j}_2).$$

Moreover,

$$v_2(T_2 \setminus \tilde{j}_1 \cup \{\tilde{j}_2\}) \geq v_2(\tilde{j}_2) \geq v_2(\hat{j}_2).$$

Thus,

$$v_2(X_2) = v_2(T_2 \setminus \tilde{j}_1 \cup \{\tilde{j}_2\}) \geq v_2(\tilde{S}_{i_2})/2 = \mu_2/2,$$

as desired.

It is left to show that $v_1(X_1) \geq \mu_1/(2 + \epsilon)$. If $i_1 = 2$, then

$$v_1(\tilde{S}_{i_1}) = v_1(\tilde{S}_2) \geq v_1(S_2) \geq (1 - \epsilon/4)\mu_1.$$

In this case, the same exact arguments used for agent 2 can be harnessed to show that $v_1(X_1) \geq (1 - \epsilon/4)\mu_1/2 \geq \mu_1/(2 + \epsilon)$. Thus, it is left to consider the case where $i_1 = 1$.

Consider the $(S_1, S_2)$ partition that is set in the first step of Cut-and-Balance-and-Choose. Since $v_1(S') \leq v_1(S_1)/2$, we have

$$v_1(\tilde{S}_1) \geq v_1(S_1)/2 \geq (1 - \epsilon/4)\mu_1/2 \geq \frac{\mu_1}{2 + \epsilon}.$$

If $\tilde{j}_2 = \hat{j}_2$, we have that

$$v_1(X_1) = v_1(\tilde{S}_1 \cup \{\tilde{j}_1\} \setminus \{\hat{j}_1\}) \geq v_1(\tilde{S}_1) \geq \frac{\mu_1}{2 + \epsilon},$$

where the first inequality follows since $v_1(\tilde{j}_1) \geq v_1(\hat{j}_1)$.

Note also that if $|S_2| = 1$ i.e., $S_2 = \{a\}$, if $\hat{j}_2 \neq a$ then $v_1(X_1) \geq v_1(S_2)$ since $a \in T_2$, similarly if $\tilde{j}_2 \neq a$ then $v_1(X_1) \geq v_1(S_2)$, finally we have $\hat{j}_2 = k_2 = a$ and $v_1(X_1) \geq \mu_1/(2 + \epsilon)$ an in the first case.

Therefore, we assume $\tilde{j}_2 \neq \hat{j}_2$ and $|S_2| > 1$, and let $\hat{j}_1' \in \arg\max_{j \in \tilde{S}_1 \setminus \{\hat{j}_1\}} v_1(j)$

$$
\begin{aligned}
v_1(X_1) &= v_1(T_1 \cup \{\tilde{j}_1\} \setminus \{\tilde{j}_2\}) \\
&= v_1(T_1) + v_1(\tilde{j}_1) - v_1(k_2) \\
&\geq v_1(\tilde{S}_1 \cup \{\hat{j}_2\} \setminus \{\hat{j}_1\}) + v_1(\hat{j}_1) - v_1(\tilde{j}_2) \\
&\geq v_1(\tilde{S}_1 \setminus \{\hat{j}_1\}) + v_1(\hat{j}_1) - v_1(\tilde{j}_2) \\
&= v_1(S_1 \setminus S' \setminus \{\hat{j}_1\}) + v_1(\hat{j}_1) - v_1(\tilde{j}_2) \\
&\geq v_1(S_1 \setminus S' \setminus \{\hat{j}_1\}) + v_1(\hat{j}_1) - v_1(\hat{j}_1') \\
&= v_1(S_1 \setminus S' \setminus \{\hat{j}_1, \hat{j}_1'\}) + v_1(\hat{j}_1),
\end{aligned}
$$

where the first inequality is since, $v_{1\tilde{j}_1} = \max_{j \in T_2} v_{1j} \geq v_{1\hat{j}_1}$. The second inequality is since $v_1(\hat{j}_2) \geq 0$, the third inequality is by $\hat{j}_1'$ definition since $\tilde{j}_2 \in \tilde{S}_1 \setminus \hat{j}_1$ by our assumption that $k_2 \neq \hat{j}_2$. Finally, we have have $|S_1 \setminus S' \setminus \{\hat{j}_1, \hat{j}_1'\}| \geq |S'|$ since

$$|S_1| - 2 - |S'| = |S_1| - 2 - (m/2 - |S_2|) = |S_1| - 2 - (m/2 - (m - |S_1|)) = m/2 - 2 \geq |S'|,$$

where the last inequality is since $|S_2| > 1$. Since we handles the case $|S_2| = 1$ earlier, we can here assume $|S_2| > 1$ in which case the set $S'$ is required to satisfy $v_1(S') \leq v_1(S_1 \setminus \{\hat{j}, \hat{j}'\})/2$. Therefore, we have $v_1(S_1 \setminus S' \setminus \{\hat{j}_1, \hat{j}_1'\}) \geq v_1(S')$.

$$
\begin{aligned}
(1 - \epsilon/4)\mu_1 \leq v_1(S_1) &= v_1(S_1 \setminus S' \setminus \{\hat{j}_1, \hat{j}_1'\}) + v_1(\hat{j}_1) + v_1(\hat{j}_1') + v_1(S') \\
&\leq v_1(S_1 \setminus S' \setminus \{\hat{j}_1, \hat{j}_1'\}) + 2 \cdot v_1(\hat{j}_1) + v_1(S') \\
&\leq 2 \cdot v_1(S_1 \setminus S' \setminus \{\hat{j}_1, \hat{j}_1'\}) + 2 \cdot v_1(\hat{j}_1) \\
&\leq 2 \cdot v_1(X_1),
\end{aligned}
$$

which implies that $v_1(X_1) \geq \frac{\mu_1}{2 + \epsilon}$, finishing the proof.

## H   Mechanisms for $n$ agents

In this section we provide a learning-augmented mechanism for $n > 2$ agents, Learning-Augmented-MMS-for-$n$-Agents. The mechanism we devise ensures that if the predictions are accurate, then each agent gets an allocation with value at least $\mu_i^n/2$ (2 consistency). On the other hand, we show that for any prediction, every agent gets at least $\mu_i^{\hat{n}}/\alpha$ for $\hat{n} = \lceil 3n/2 \rceil$ and $\alpha = \max\{m - \hat{n} - 1, 1\}$ (robustness).

**Theorem H.1.** *The Learning-Augmented-MMS-for-$n$-Agents Mechanism (Mechanism 6) is truthful, 2-consistent and $(\mu_i^{\hat{n}}/\alpha)$-robust for $\hat{n} = \lceil 3n/2 \rceil$ and $\alpha = \max\{m - \hat{n} - 1, 1\}$.*

## H.1  An Overview

**The Mechanism.**    The mechanism works in three phases. In the first phase, it uses the predictions in order to obtain a partial allocation to agents with high predicted items (which are then removed from the set of active agents, so that we can now that for all agents, all predicted values are small). Then, in the second stage, the mechanism uses the predictions in order to obtain a *tentative allocation*, by running a Round-Robin procedure, where items are tentatively allocated to agents according to their predictions. In the third and final phase, the tentative allocation is used to implement a *recursive* plant and steal procedure, where the "planting" is done from the tentative allocations according to predictions, but the "stealing" is done according to the agents' reports and results in a *final allocation*.

---

**MECHANISM 6:** `Learning-Augmented-MMS-for-`$n$`-Agents`

**Input** : Set of agents $N$, set of items $M$, reports $\mathbf{r}_N$, predictions $\mathbf{p}_N$
**Output :** A partition of the items $\bigcup_{i \in N} X_i$
Invoke Algorithm 7, $X \leftarrow$ `Allocate-Large`$(N, M, \mathbf{r}_N, \mathbf{p}_N)$
Invoke Algorithm 9, $A \leftarrow$ `Tentative-Allocation-Round-Robin`$(N, M, \mathbf{p}_N)$
Invoke Algorithm 10, $X \leftarrow$ `Split-Plant-Steal-Recurse`$(N, A, \textit{first-level-flag} = \textbf{True}, X, \mathbf{r}_N, \mathbf{p}_N)$

---

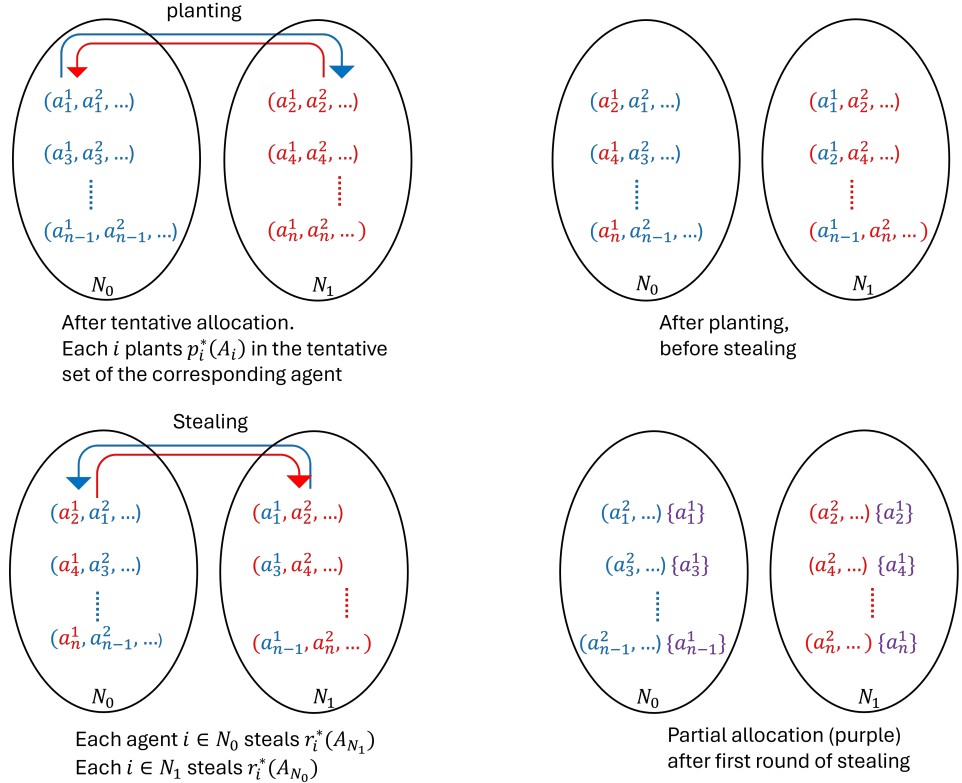

Figure 3: Illustration of a single round of the recursive planting and stealing phase (Algorithm 10), for the case where predictions are accurate (so that each agent steals back their planted item). Note that the stealing is done from the union of items of agents in the opposite set (and not just from the corresponding agent).

**Consistency.**    In the case the predictions are accurate, the initial allocation phase will take care of agents with high valued items (of value larger than $\mu_i^n/2$). Then, in the second phase, the tentative allocation will be exactly identical to a Round-Robin allocation (made according to true valuations). Finally, in the third phase, since agents steal in the same order they were allocated the items in the Round-Robin allocation, and since the predictions are accurate, the agents "steal" back the same item the mechanism plants. Since a Round-Robin allocation achieves $\mu_i^n/2$ when there are no agents with high valued items [8], correctness follows.

**Robustness.** In the case the predictions are inaccurate, we show that every agent still gets at least $\mu_i^{\hat{n}}/\alpha$. Here we rely on the plant-and-steal phase to ensure that each agent gets at least their $\lceil 3n/2 \rceil$ highest-valued item according to their true valuation. This property provides our robustness guarantee. We notice that reversing the order between the first and subsequent rounds of the Round-Robin procedure (and thus, the stealing phases) gives an enhanced robustness guarantee.

**Prediction.** In the description of the mechanism, we assume the mechanism is given a prediction of agents valuations. We note that in order to implement the mechanism it is enough to be given access to agents' preference order over items, and an additional information indicating which items are worth more than $\mu_i^n/2$ for each agent $i$.

Due to space constraints the proof of Theorem H.1 is deferred to Appendix H.3, and we now provide a detailed description of the different phases.

## H.2 Implementation Details

As discussed, in order to utilize the Round-Robin mechanism, we first allocate a single item to each agent with a high predicted value.

---

**ALGORITHM 7:** `Allocate-Large`

---

**Input** : Set of agents $N$, set of items $M$, reports $\mathbf{r}_N$, predictions $\mathbf{p}_N$
**Output** : A partial allocation $\bigcup_{i \in B} X_i$, updated sets of agents and items $N, M$, respectively
**foreach** $i \in N$ **do**
    Compute $\mu_i^n$ based on $p_i$
**while** *exists* $i \in N$ *such that* $p_i^*(M) \geq \mu_i^n/2$ **do**
    $X_i \leftarrow \{r_i^1(M)\}$
    $M \leftarrow M \setminus X_i$
    $N \leftarrow N \setminus \{i\}$

---

Before describing the tentative allocation mechanism, we first give a procedure, `Allocate-Best`, which performs a single round of Round-Robin according to a specific order, and preferences (either predictions or reports), denote $\mathbf{o}$.

---

**PROCEDURE 8:** `Allocate-Best` (One-Round-RR)

---

**Input** : Ordered set of agents $N$, set of items $M$, valuation $\mathbf{v}_N$
**Output** : $|N|$ singletons $X_i \in M$
**foreach** $i \in N$ **do**
    $X_i \leftarrow v_i^*(M)$
    $M \leftarrow M \setminus X_i$

---

The tentative allocation mechanism repeatedly invokes `Allocate-Best` according to given predictions, until all items are tentatively allocated. As previously mentioned, the first round of the tentative allocation is performed according to the given order, and in all subsequent rounds, the order is reversed (recall that reversing the order enhances the robustness guarantees).

---

**ALGORITHM 9:** `Tentative-Allocation-Round-Robin`

---

**Input** : Ordered set of agents $N = (i_1, \ldots, i_{|N|})$, set of items $M$, predictions $\mathbf{p}_N$
**Output** : A tentative allocation $\bigcup_{i \in N} A_i = M$
$A \leftarrow$ `Allocate-Best`$(N, M, \mathbf{p}_N)$
$M \leftarrow M \setminus \cup_{i \in N} A_i$

      /* Reverse the order for the allocation of the rest of the items */
$N^r = (i_{|N|}, \ldots, i_1)$
**for** $k = 2, \ldots, \lceil m/n \rceil$ **do**
    $\tilde{A} =$ `Allocate-Best`$(N^r, M, \mathbf{p}_{N^r})$
    $A_i \leftarrow A_i \cup \tilde{A}_i$ for $i \in N$
    $M \leftarrow M \setminus \cup_{i \in N} \tilde{A}_i$

---

The final phase in the mechanism is a recursive plant and steal algorithm. The input to this algorithm is an ordered set of agents $N$, along with their predictions, reports, and a tentative allocation for each agent. At each recursive invocation, the algorithm splits the set of agents into two (almost) equal-size ordered sets $N_0$ and $N_1$. Then the mechanism "plants" for the $i^{\text{th}}$ agent in each set $N_b$ their highest (according to predictions) valued item in their tentative allocation in the tentative set of the $i^{\text{th}}$ agent in $N_{\neg b}$. Then we perform one round of Round-Robin, where the items available to the agents of set $N_b$ are those tentatively allocated to the agents of $N_{\neg b}$ (after the planting phase), and the allocations are determined according to agents reports. See Figure 3 for an illustration of a single round of plant and steal. The algorithm then recurses on each of the sets $N_0$ and $N_1$, until all sets are of size 1. At this point, the single agent in the set is further allocated its remaining tentatively allocated items, and the process terminates.

---

**ALGORITHM 10:** `Split-Plant-Steal-Recurse`

---

**Input** : Ordered set of agents $N = (i_1, \ldots, i_{|N|})$, tentative allocations $A$, partial allocations $X_N$, *first-level-flag* indicating if this is the first level of the recursion, reports $\mathbf{r}_N$, predictions $\mathbf{p}_N$

                  /* Halting condition - Allocate all remaining items */
**if** $N = \{i\}$ **then** set $X_i = X_i \cup A_i$ and *halt*

                  /* Split the agents into two almost-equal parts */
$par = |N| \mod 2$
$N_0 \leftarrow (i_1, i_3, \ldots, i_{|N|-1+par})$
$N_1 \leftarrow (i_2, i_4, \ldots, i_{|N|-par})$

                  /* Plant according to predictions */
**for** $i = 1, \ldots, \lfloor |N|/2 \rfloor$ **do**
    Let $i^0, i^1$ denote the $i^{\text{th}}$ agent in $N_0, N_1$ respectively.
    $j_0^* = p_{i^0}^*(A_{i_0})$
    $j_1^* = p_{i^1}^*(A_{i_1})$
    $A_{i^0} = A_{i^0} + j_1^* - j_0^*$
    $A_{i^1} = A_{i^1} + j_0^* - j_1^*$

                  /* Plant $i_n$'s favorite item in a tentative set */
**if** $par = 1$ **then**
    $i^0 = i_n, i^1 = i_2$
    $j_0^* = p_{i^0}^*(A_{i^0})$
    $A_{i^1} = A_{i^1} + j_0^*, A_{i^0} = A_{i^0} - j_0^*$

                  /* Steal from the opposite set according to reports */
**foreach** $b \in \{0, 1\}$ **do**
    $\hat{X}$=Allocate-Best$(N_b, A_{N_{\neg b}}, \mathbf{r})$
**foreach** $i \in N$ **do**
    $X_i \leftarrow X_i \cup \hat{X}_i$

                  /* Reverse the order after the first level of recursion */
**if** *first-level-flag* **then**
    $N_0 \leftarrow (i_{|N|-1+par}, \ldots, i_3, i_1)$
    $N_1 \leftarrow (i_{|N|-par}, \ldots i_4, i_2)$

                  /* Recursively invoke Split-Plant-Steal-Recurse on each set */
**foreach** $b \in \{0, 1\}$ **do**
    Split-Plant-Steal-Recurse$(N_b, A_{N_b}, X_{N_b}, \textit{first-level-flag} = \textbf{False})$

---

Given the above implementation details, it remains to prove Theorem H.1 regarding truthfulness, consistency and robustness of the mechanism. The proof is given in Appendix H.3.

## H.3 Missing Details from Section H

In this section we prove Theorem H.1, which we now recall.

**Theorem H.1.** *The* `Learning-Augmented-MMS-for-`$n$`-Agents` *Mechanism (Mechanism 6) is truthful, 2-consistent and* $(\mu_i^{\hat{n}}/\alpha)$*-robust for* $\hat{n} = \lceil 3n/2 \rceil$ *and* $\alpha = \max\{m - \hat{n} - 1, 1\}$.

First, we give a simple observation regarding Algorithm 7.

**Observation H.1.** *The followings hold for Algorithm* `Allocate-Large`.

1. *If the reports equal the true valuations, and agent $i$ is allocated an item $j$, then $v_i(j) \geq v_i^n/2$.*

2. *After the algorithm completes its run, there are no remaining agents in $N$ with large predicted values for the remaining items in $M$.*

We continue to prove each of the properties specified in Theorem H.1 separately, starting with truthfulness.

**Lemma H.1** (Truthfulness). *Mechanism* `Learning-Augmented-MMS-for-$n$-agents` *(Mechanism 6) is truthful.*

*Proof.* Algorithm `Tentative-Allocation-Round-Robin` (Algorithm 9) only depends on agents predictions and not their reports. Hence, we only need to consider the use of the reports in Algorithms 9 and 10.

For every agent $i$, either they are allocated a single item in Algorithms 9, or $i$ participates in the recursive plant ant steal, and this is determined according to the predictions, so in particular $r_i$ has no affect on this. Thus, we can consider the two independent events separately. In the first case, where $i$ is allocated a single item, it is the item that maximizes their report over remaining items at that point, so that $i$ has no incentive to lie.

In the second case, $i$ participates in the plant and steal phase. Observe that in this case, whenever $i$ chooses an item from some set $A'$, it will have no future interaction with this set. That is, fix a recursive call and assume without loss of generality that $i \in N_0$. Then after the planting step, $i$ is allocated the item in $A_{N_1}$ that maximizes their reports. Then, in following recursive steps, $i$ only continues to interact with items in $A_{N_0}$, so $i$'s choice does not affect the identity of the items from which $i$ will be able to choose from in future rounds. Hence, $i$'s only incentive is to maximize the value of its allocated value in each round, implying truthfulness. $\square$

Due to the above lemma, from now on we assume agents report truthfully, i.e., that for every agent $i$, $r_i = v_i$. We turn to show the mechanism is consistent, we rely on the following theorem.

**Theorem H.2** (Lemma 2 in [10] (based on Theorem 3.5 in [8])). *If for every $i \in N$ and $j \in M$, $v_i(j) \leq \frac{1}{2}\mu_i^n$, then the Round-Robin algorithm returns an allocation that is a 2-approximation to the MMS.*

*Furthermore, their analysis holds when changing the order of allocation between the different rounds of the Round-Robin.*

We are now ready to prove the mechanism is consistent.

**Lemma H.2** (Consistency). *If the set of predictions is accurate, then for every $i$, $v_i(X_i) \geq \mu_i^n/2$.*

*Proof.* First consider agents that were allocated an item in Algorithm `Allocate-Large` (Algorithm 7). If the predictions are accurate, then each such agent $i$ is allocated an item $j$ such that $v_i(j) \geq \mu_i^n/2$ and so the statement holds. Moreover, at the end of this step, there are no remaining agents with large predicted values, hence, no agents with large values remain.

If the set of predictions is accurate, then the tentative allocation determined according to agents' predictions in Algorithm `Tentative-Allocation-Round-Robin` ( Algorithm 9) is identical to a Round-Robin mechanism according to valuations, with reversing the order between the first and all subsequent rounds. Furthermore, by the above, there are no agents with large values when the Round-Robin is invoked. Therefore, by Theorem H.2, it holds that for every $i$, $v_i(A_i) \geq \mu_i^n/2$. We shall prove that for every agent $i$, its final allocation equals its tentative allocation, $X_i = A_i$, concluding the proof.

We prove that in depth $k$ of the recursion, every agent $i$ is allocated the $k^{th}$ item in $A_i$. We prove the claim by induction on the depth $k$ of the recursion, and the $\ell^{th}$ agent in that round that is allocated some value.

We first prove for $k = 1$, $\ell = 1$. In the plant phase, $\ell^0 (= 1)$ plants $j = p_{\ell^0}^*(A_{\ell^0})$ in $A_{\ell^1}$. Then, in the stealing phase, during the invocation of Algorithm 8, agent $\ell^0$ is the first to choose an item from $A_{N_1}$,

which in particular contains $j$. Hence, the first item in $A_1$ is allocated into $X_1$. We now assume the claim holds for $k = 1$ and $\ell - 1$ and prove it for $\ell$. Assume without loss of generality that $\ell$ is odd so that $i_\ell \in N_0$.

In step $\ell$ of the planting phase, the mechanism plants $\ell^0$'s (the proof for $\ell^1$ is identical) first (according to value $p_{\ell^0}$) item in $A_{\ell^0}$. Then, during the tentative allocation phase, agent $\ell^0$ is the $\ell^{\text{th}}$ to choose among the items in $A_{N_1}$ minus the items that were allocated to the $\ell - 1$ agents that were before her in the tentative Round-Robin. By the induction hypothesis, every agent preceding her chose the item the mechanism planted for them previously in that round. Therefore, the item $j$ that the mechanism planted for agent $\ell^0$ is still available. Moreover, let $M^{\ell-1}$ denote the set of items after $\ell - 1$ rounds of the tentative Round-Robin in Algorithm 9. Further let $A_{N_1}^{\ell-1}$ denote the set of items after $\ell - 1$ rounds of the `Allocate-Best` algorithm invoked in the stealing phase with the set $N_0$, i.e., $A_{N_1}^{\ell-1} = A_{N_1} \setminus \bigcup_{j \in N_0, j < \ell} \{X_j\}$. Since the order in which the agents plant and steal in each round of the recursion is equivalent to the order in which the corresponding tentative allocation round was performed, it holds that $A_{N_1}^{\ell-1} \subset M^{\ell-1}$. Since $j = p_{\ell^0}^*(M^{\ell-1})$, and $p_{\ell^0} = r_{\ell^0}$, it holds that $r_{\ell^0}^*(A_{N_1}^{\ell-1})$ equals $j$. Therefore $\ell^0$ will choose $j$ to $X_{\ell^0}$ as claimed.

Proving the claim for a general $k$ is almost identical. At the planting phase of the $k^{\text{th}}$ round, the mechanism plants for every agent $\ell^0 \in N_0^k$ their $k^{\text{th}}$ item of $A_i$ in $A_{N_1^k}$ and vice versa. A similar argument to the one above, shows that this item will remain available until its their turn to choose an item for allocation, as by the recursion hypothesis, all agents preceding $i$ in the Round-Robin will select the items the mechanism planted for them. Hence, the $k^{\text{th}}$ item in $A_{\ell^0}$ will be allocated to $X_{\ell^0}$.

Finally, once the set agent $i$ belongs to becomes a singleton, by our halting condition, $X_i \leftarrow X_i \cup A_i$, so together with the previous argument, we get that for every $\ell$, $X_i = A_i$ as needed. $\qquad \square$

We continue to prove that the mechanism is robust. Since when $m < \hat{n}$, $\mu_i^{\hat{n}} = 0$ for every agent $i$, and each agent trivially gets their MMS value, we assume from now on that $m \geq \hat{n}$ and show the mechanism achieves $(m - \hat{n} - 1)$-robustness for $\mu_i^{\hat{n}}$. We first prove in Lemma H.3 that for each agent $i$, $v_i(X_i) \geq v_i^{\lceil 3n/2 \rceil}$, and then prove in Lemma H.5 that the value of this item is not too small compared to $\mu_i^{\lceil 3n/2 \rceil}$.

**Lemma H.3.** *For every agent $i$, $v_i(X_i) \geq v_i^{\lceil 3n/2 \rceil}$.*

*Proof.* We first prove the claim for agents that were allocated a value during the invocation of Algorithm 7. By the definition of the algorithm and its truthfulness when agent $i$ is allocated an item, at most $n - 1$ items were previously allocated to other agents. Hence, she can always choose her $n^{\text{th}}$ highest valued item. Therefore, we have $v_i(X_i) \geq v_i^n \geq v_i^{\lceil 3n/2 \rceil}$, as claimed.

We continue to prove the claim for the set of agents with no large predicted values. Consider the $\ell^{\text{th}}$ agent in $N$, $i_\ell$, and consider the following coloring process. Initially, color all items in $M$ black. We will then color all items $i_\ell$ was able to choose from *green*, and items allocated before she had the chance to choose from *gray* (note that these colors are unrelated to the ones in the figure). Note that an item turns green when it belongs to the tentative allocation of opposite set to $i_\ell$'s and has not been taken by agents preceding her in the allocation order. We claim that by the time no black items remain, at most $\lceil 3n/2 \rceil - 1$ have turned gray, implying that at some point during the recursion, $i_\ell$ could have chosen their $\lceil \frac{3n}{2} \rceil^{\text{th}}$ highest valued item (according to $r_{i_\ell}$).

We let $N^k$ denote the set of agents to which $i_\ell$ belongs to at depth $k$ of the recursion, starting with $N^1 = N$. At each recursive call, $N^k$ is partitioned into $N_0^k, N_1^k$. We further let $b^k \in \{0, 1\}$ denote the index of the set to which $i_\ell$ belongs to: $i_\ell \in N_{b^k}^k$. We will separately bound the number of items turned gray due to agents in $N_{b_k}^k$ and $N_{\neg b_k}^k$.

In the first iteration, for $k = 1$, let $A_{N_{b^0}^1}^1, A_{N_{\neg b^0}^1}^1$ denote the tentative sets allocated to the agents of $N_0^1$ and $N_1^1$ after the planting phase (i.e., at the beginning of the stealing phase).

The number of items that turn gray due to agents in $N_{b^1}^1$ is $G_{b^1}^1 = \lceil \ell/2 \rceil - 1$, since $i_\ell$ has access to all items in $A_{N_{\neg b^1}}^1$ excluding the $\lceil \ell/2 \rceil - 1$ items that were allocated to the agents in her set preceding her in the ordering. (The rest of the items in $A_{N_{\neg b^1}}^1$ turn green.)

Turning to $G^1_{\neg b^1}$, each agent in the opposite set to hers, $N^1_{\neg b^1}$, is allocated a single item (from $A^1_{N^1_{b^1}}$) before continuing to the next round of the recursion. Therefore, $G^1_{\neg b^1} = |N^1_{\neg b^1}|$ (and no item turns green).

The recursion then continues with $N^2 = N^1_{b^1}$ and in reversed order (due to the order being reversed). Therefore, at the beginning of the second iteration, $i_\ell$ is in location $|N^1_{b^1}| - \lceil \ell/2 \rceil$ in $N^2$. After the partition phase, $i_\ell$ is in set $N^2_{b^2}$ and in location $\lceil \frac{|N^1_{b^1}| - \lceil \frac{\ell}{2} \rceil}{2} \rceil$. Hence, $G^1_{b^1} = \lceil \frac{|N^1_{b^1}| - \lceil \frac{\ell}{2} \rceil}{2} \rceil - 1$ due to agents in her set preceding here in the ordering. Also, $G^1_{\neg b^1} = |N^1_{\neg b^1}|$ due to allocations to agents in the opposite set to hers.

From now on, the order is preserved, so for every $k \geq 3$, $G^k_{b^k} = |N^k_{b^k}|$ and $G^k_{\neg b^k} = \lceil \frac{|N^1_{b^1}| - \lceil \ell/2 \rceil}{2^{k-1}} \rceil - 1$.

We continue by bounding $\sum_{k=1}^{\lceil \log n \rceil} G^k_{\neg b^k} = \sum_{k=1}^{\lceil \log n \rceil} |N^k_{\neg b^k}|$. Observe that if $N^k$ is even then $N^k_{b^k} = N^k_{\neg b^k} = N^k/2$, and if $N^k$ is odd, then either $N^k_{b^k}$ is odd and $N^k_{\neg b^k}$ is even or vice versa. In the first case, $G^k_{\neg b^k} = \lceil N^k/2 \rceil$ and we recurse with $N^k_{b^k}$ which is of size $\lfloor N^K/2 \rfloor$. In the second case, $G^k_{\neg b^k} = \lfloor N^K/2 \rfloor$ and we recurse with $N^k_{b^k}$ of size $\lceil N^k/2 \rceil$. Hence, we have the following recursion formula. For even $\ell$, $T(\ell) = \ell/2 + T(\ell/2)$, and for odd $\ell$, either (a) $T(\ell) = \lceil \ell/2 \rceil + T(\lfloor \ell/2 \rfloor)$ or (b) $T(\ell) = \lfloor \ell/2 \rfloor + T(\lceil \ell/2 \rceil)$. In Claim H.4 below, we prove that for such a function, if it also holds that $T(1) = 0$ and $T(2) = 1$, then $T(\ell) \leq \ell - 1$. Therefore, we get that $\sum_{k=1}^{\lceil \log n \rceil} G^k_{\neg b^k} \leq n - 1$.

We continue to bound $\sum_{k=2}^{\lceil \log n \rceil} G^k_{\neg b^k} = \sum_{k=2}^{\lceil \log n \rceil} \lceil \frac{|N^1_{b^1}| - \lceil \ell/2 \rceil}{2^{k-1}} \rceil - 1$. The sum $\sum_{k=1}^{\lceil \log X \rceil} \lceil \frac{X}{2^k} \rceil$ can be bounded by $\left( \sum_{k=1}^{\lceil \log X \rceil} \frac{X}{2^k} \right) + L$, where $L$ is the number of indices $k$ for which the fraction $X/2^k$ is rounded up. Observe that for every $X$, $L$ can be bounded above by $\lceil \log X \rceil$ as $L$ exactly equals the number of 1 bits in the binary representation of $X$. Hence, the overall number of items that turn gray can be bounded as follows:

$$
\begin{aligned}
G^{\lceil \log n \rceil} &= \sum_{k=1}^{\lceil \log n \rceil} \left( G^k_{\neg b^k} + G^k_{b^k} \right) \\
&\leq n - 1 + \lceil \ell/2 \rceil - 1 + \sum_{k=2}^{\lceil \log n \rceil} \left( \left\lceil \frac{\lceil n/2 \rceil - \lceil \ell/2 \rceil}{2^{k-1}} \right\rceil - 1 \right) \\
&\leq n - 1 + \lceil \ell/2 \rceil - 1 + \lceil n/2 \rceil - \lceil \ell/2 \rceil + \lceil \log n \rceil - \lceil \log n \rceil + 1 \\
&\leq \lceil 3n/2 \rceil - 1.
\end{aligned}
$$

Therefore, the number of items that turn gray by the end of the recursion is at most $\lceil 3n/2 \rceil - 1$, and so $i_\ell$ get their $\lceil 3n/2 \rceil$ highest valued item $v_{i_\ell}^{\lceil 3n/2 \rceil}$. □

We now prove the claim regarding the cost of the recursion that was used in the previous lemma.

**Lemma H.4.** *Let $T(n)$ be such that $T(n) = n/2 + T(n/2)$ if $n$ is even and either (a) $T(n) = \lceil n/2 \rceil + T(\lfloor n/2 \rfloor)$ or (b) $T(n) = \lfloor n/2 \rfloor + T(\lceil n/2 \rceil)$ for odd $n$. Also assume $T(1) = 0, T(2) = 1$. Then $T(n) \leq n - 1$.*

*Proof.* We prove the claim by induction on $n$. By $T(1) = 0$ and $T(2) = 1$ so the induction basis holds. We now assume correctness for all values smaller than $n$ and prove for $n$.

If $n$ is even then $T(n) = n/2 + T(n/2) \leq n/2 + n/2 - 1 = n - 1$, so the claim holds.

If $n$ is odd, then in case (a), $T(n) = \lceil n/2 \rceil + T(\lfloor n/2 \rfloor) \leq \lceil n/2 \rceil + \lfloor n/2 \rfloor - 1 = n - 1$, and in case (b), $T(n) = \lfloor n/2 \rfloor + T(\lceil n/2 \rceil) - 1 \leq \lfloor n/2 \rfloor + \lceil n/2 \rceil - 1 = n - 1$. □

Finally, we prove that the highest valued item allocated to each agent $i$ is not too small compared to their MMS.

**Lemma H.5.** *Consider an MMS for agent $i$, and let $j^*$ be the highest valued item of $i$ in her allocation. Then*

$$
v_i(j^*) \geq \mu_i^{\lceil 3n/2 \rceil} / \alpha \quad \text{for} \quad \alpha = m - \lceil 3n/2 \rceil - 1.
$$

*Proof.* Consider an MMS allocation of $M$ for $k = \lceil 3n/2 \rceil$, and let $A_i$ be the set such that $v_i(A_i) = \mu_i^k$. By the assumption on $j^*$, its value is higher then the highest valued item in $A_i$, $v_i(j^*) \geq v_i^1(A_i)$. Therefore, $v_i(A_i) \leq |A_i| \cdot v_i(j^*)$, implying $v_i(j^*) \geq v_i(A_i)/|A_i| = \mu_i^k/|A_i|$. Since $|A_i| \leq m - k - 1$ (as at least $k - 1$ items must be allocated to the $k - 1$ additional agents, it holds that $v_i(j^*) \geq \mu_i^{3n/2}/(m - \lceil 3n/2 \rceil - 1)$. $\qquad\square$

*Proof of Theorem H.1.* The theorem follows by Lemmas H.1, H.2, H.3, and H.5. $\qquad\square$

