# OpenReview forum: "Plant-and-Steal: Truthful Fair Allocations via Predictions"
_NeurIPS.cc/2024/Conference — NeurIPS 2024 poster_

### Official Review · Reviewer_CcvG · 2024-06-12

**Soundness:** 3
**Presentation:** 3
**Contribution:** 3
**Rating:** 6
**Confidence:** 3

**Summary:**

This paper considers the problem of fairly and truthfully allocating indivisible items where mechanisms are equipped with predictions. When the prediction is perfectly accurate, the mechanism's performance should be significantly improved (consistency); conversely, for a prediction with any accuracy, the mechanism's performance should still be guaranteed (robustness). The paper aims to design truthful mechanisms with predictions that optimize the approximation to maximin share (MMS).

The paper achieves a variety of consistency-robustness trade-offs under different settings. For $2$ agents and ordering predictions, which predict agents' ranking preferences over items, the paper gives a truthful mechanism that is $2$-consistent and $\lceil m/2 \rceil$-robust, where the robustness guarantee matches the best approximation ratio achievable by truthful mechanisms without predictions. Another truthful mechanism is provided with $3/2$-consistency and $\lfloor 2m/3 \rfloor$-robustness.

For $2$ agents and arbitrary predictions, the paper gives a lower bound for consistency when the robustness is bounded. Moreover, the paper studies the trade-offs achievable by more space-efficient predictions. Finally, for any number of agents, the paper presents a truthful mechanism with $2$-consistency and a slightly relaxed robustness guarantee.

**Strengths:**

(1) The paper studies an important problem. Given that strong impossibility results exist when facing strategic agents, it's natural to bridge the gap between the strategic and non-strategic settings via predictions.

(2) The paper is well-written and carefully structured.

(3) The results are interesting, and the techniques are non-trivial. In particular, for two agents, the approximation ratios match the best achievable bounds up to a constant factor when the prediction is perfectly correct or completely incorrect.

**Weaknesses:**

(1) The motivation for studying space-efficient predictions is not convincing enough.

(2) Most of the results only hold for two agents.

(3) The lower bound is not very promising as even in the non-strategic setting, the best approximation ratio for MMS is larger than $4/3$.

(4) Minor:
- Line 28: "probability 2"
- Line 60: Should briefly explain what it means by "ex-post" guarantees.
- Line 121: $\mu_i$ is not defined before.
- Line 212: "over agents items"
- Line 228: "according predictions"
- Line 254: "present present"
- Line 271: "show prove prove"
- Line 291: "thenB"
- Line 320: "to allocated"

**Questions:**

Can you further explain why studying space-efficient predictions is important and interesting?

**Limitations:**

Some of the results are not sufficiently motivated or not promising.

---

> ### Author Rebuttal · Authors · 2024-08-07
>
> "Can you further explain why studying space-efficient predictions is important and interesting?"
>
> -- Our motivation for succinct predictions comes from the works of [5,6,7], where they show that
> succinct predictions are crucial for learning the parameters from a few samples and for incorporating a PAC-learnable component in the learning-augmented framework under plausible distributional assumptions. We believe that combining our results on small space predictions with an adequate distributional assumption will yield such a result. We plan to formalize this in subsequent work.
>
> [5] Ilan Reuven Cohen and Debmalya Panigrahi. A General Framework for Learning-Augmented Online Allocation, ICALP 2023.
>
> [6] T Lavastida, B Moseley, R Ravi, and C Xu. Learnable and instance-robust predictions for online matching, flows and load balancing, ESA 2021.
>
> [7] Shi Li and Jiayi Xian. Online unrelated machine load balancing with predictions revisited, ICML 2021.

---

> > ### Comment · Reviewer_CcvG · 2024-08-09
> >
> > Thanks for addressing my concern, and please add the explanations to the paper. I will maintain my score.

---

### Official Review · Reviewer_iNp4 · 2024-07-05

**Soundness:** 3
**Presentation:** 3
**Contribution:** 2
**Rating:** 4
**Confidence:** 3

**Summary:**

The authors study the problem of fairly allocating a set of $m$ indivisible goods among a set of $n$ strategic agents with additive valuations in a fair manner. The goal is to obtain a truthful mechanism which guarantees a good approximation of maximin share (MMS) of each agent. It has already been shown that no truthful mechanism can guarantee better than $\lfloor \frac{2}{m} \rfloor$-MMS, while a $\frac{720}{959}>\frac{3}{4}$ MMS allocations can be guaranteed when the agents' valuations are public. This paper bridges this gap by having a learning augmented lens and using predictions on agents' valuations.

In particular, the authors introduce a framework called "plant-and-steal" which works as follows. For two agents, given a prediction $p$ and an algorithm $A$, it runs $A$ on $p$ (i.e., assuming the valuations are as given in $p$). Then from the bundle of agent $i$, a most valuable good based on the prediction $p$ is taken from $i$'s bundle and given (planted) in the other agents bundle. This is done at the same time for both agents. Then, both agents steal a most valuable good based on their reported values from the other agents bundle. It is easy to see that this mechanism is truthful. The authors prove that if the prediction $p$ predicts the ordering of the goods for the agents and the algorithm $A$ is the round-robin algorithm, then plant-and-steal is $1/2$-consistent and $\lceil \frac{2}{m} \rceil$-robust. Furthermore, they prove that the guarantee "gracefully" degrades in the number of mistakes in the prediction. They use Kendall tau distance as a measure of preciseness of the prediction.

For more number of agent, they prove a similar approach gives $1/2$-consistency and $((m - \lceil 3n/2 \rceil - 1)^{-1}, \lceil 3n/2 \rceil )$-robustness. It means that no matter how bad the predictions, at least $(m - \lceil 3n/2 \rceil - 1)^{-1}$ fraction of the MMS values of each agent is guaranteed assuming that the number of agents is $\lceil 3n/2 \rceil$.

Finally, they also have experiments showing how well their framework works in practice.

**Strengths:**

The paper is well-written and easy to follow and I did enjoy reading it. Looking at fair division problems through the lens of learning augmented algorithms is definitely an interesting approach. At least for the case of two agents, the paper presents an almost optimal result that one can hope for.

**Weaknesses:**

The contribution of the paper is limited in my opinion since the main result only concerns two agents. For more number of agents, while the consistency guarantee is not bad, the robustness guarantee is very weak.

I have a more fundamental concern regarding the presented algorithm. The existing algorithms for approximate MMS (in the classic fair division setting without predictions) guarantee $\alpha$-MMS allocations and currently the best known $\alpha$ is marginally above $3/4$. These algorithms are not truthful mechanisms, however, they guarantee every single agent that if she is truthful, she will end up having $\alpha$ fraction of her MMS value. In particular for two agents, $\alpha=1$. Now, what is the use of having truthful mechanism which guarantee only $2/m$ fraction of MMS values of the agents? In particular, the proposed mechanism, incentivise the agents to be truthful, and in the best case guarantees $1/2$-MMS and if its own prediction is off, even much worse. What I am trying to convey is that, while there exists algorithms that guarantee $3/4$ and better approximation of MMS, what is the incentive for strategic agents to participate in a truthful mechanism, give away all their data (since their dominant strategy is to be truthful) and in return get way less than they could possibly get.

**Questions:**

Could you please address the raised concern in the last section?

Comments and Suggestions:

L 22: For the case of two agents ... : the sentence does not read well. Maybe replace "which" with "what"

L 40: ... over goods.For ... $\rightarrow$ .. over goods. For ...

L 58: with probability $2$ $\rightarrow$ with probability $1/2$

L 57 until the end of the paragraph: I did not understand why you mentioned randomized allocations at all and if so, why so briefly. In my opinion you should either discuss it properly and cite the related work or not at all. Having only one sentence on it was a bit confusing.

L 92: $\lceil \frac{m}{2} \rceil$ $\rightarrow$ $\lfloor \frac{m}{2} \rfloor$

L 122: ran $\rightarrow$ run

L 125: then $\rightarrow$ than

L 163: use \boldmath in the paragraph title

L 209, 213: $\ell$th $\rightarrow$ $\ell$-th

L 254: present present $\rightarrow$ present

L 264: espace before "for"

L 271: show prove prove $\rightarrow$ prove

L 291: space after "then"

L 320: allocated $\rightarrow$ allocate

**Limitations:**

No. The main contribution of the paper is theoretical. However, still from theoretical point of view, there are limits that could have been discussed. For instance while the robustness guarantee (for two) agents is almost optimal, the consistency guarantee is far from optimal.

---

> ### Author Rebuttal · Authors · 2024-08-07
>
> * "I have a more fundamental concern regarding the presented algorithm. The existing algorithms for approximate MMS (in the classic fair division setting without predictions) guarantee 𝛼-MMS allocations and currently the best known 𝛼 is marginally above 3/4. These algorithms are not truthful mechanisms, however, they guarantee every single agent that if she is truthful, she will end up having 𝛼
>  fraction of her MMS value. In particular for two agents, 𝛼=1. Now, what is the use of having truthful mechanism which guarantee only 2/𝑚 fraction of MMS values of the agents? In particular, the proposed mechanism incentivises the agents to be truthful, and in the best case guarantees 1/2-MMS and if its own prediction is off, even much worse. What I am trying to convey is that, while there exists algorithms that guarantee 3/4
>  and better approximation of MMS, what is the incentive for strategic agents to participate in a truthful mechanism, give away all their data (since their dominant strategy is to be truthful) and in return get way less than they could possibly get."
>
> Mechanism design has been a fruitful and central research area in Economics. The philosophy behind this approach is that if the mechanism is not truthful, the input for the mechanism might be strategic, and not representing the real parameters of the problem, thus the algorithms won’t be optimizing the objective it’s designed to optimize. Even if agents are to get better utility by coordinating, if each agent locally optimizes their utility, both agents might end up in a suboptimal solution, as demonstrated by the prisoner’s dilemma, while the optimal solution is not stable in a game-theoretic sense. The tension between truthfulness and the performance of the algorithm has been a central theme in the Algorithmic Game Theory literature, where the much desired truthfulness property often comes at the expense of the algorithm’s performance. Like the reviewer, we also were not satisfied with the $\lfloor m/2\rfloor$ lower bound from [2]. We also believe that many of the relevant settings for fair division, such as course allocation, data is abundant, and can be used to improve the performance of the truthful mechanism. We believe that this point was thoroughly demonstrated by our comprehensive set of results.
>
> While the privacy concern expressed by the reviewer is not the focus of our paper, we note that our mechanisms only require agents to minimally expose information, as they are only required to choose a single item from a predetermined set of items, which does not depend on their reports. Thus, they only need to reveal which is the most valuable item from a set of items, while not even exposing their value for the item. We will stress this point in the final version of the paper.
>
> [2] G. Amanatidis, G. Birmpas, G. Christodoulou, and E. Markakis. Truthful allocation mechanisms without payments: Characterization and implications on fairness, EC 2017.

---

> > ### Comment · Reviewer_iNp4 · 2024-08-10
> >
> > I thank the authors for their response. Let me describe my concern in more detail. I understand the motivation behind having truthful mechanisms in general. However, I believe the known results on MMS (unfortunately) lessen the significance of this study. It is already known that $\alpha$-MMS allocations exist for some $\alpha>3/4$. Let's assume this algorithm does not have access to all the data but the agents report their valuations. This algorithm guarantees each agent $i$, $\alpha MMS_{v'_i}$ assuming that agent $i$ reports $v'_i$. This is completely independent of what other agents are reporting. Hence, for this algorithm to output an $\alpha$-MMS allocation, it does not need all the agents to report truthfully. If someone can gain more by misreporting, it does not make other agents to end up with less than $\alpha$ fraction of their MMS value as long as they report truthfully (which makes it different from the prisoner's dilemma). On the other end, it has also been shown that no truthful mechanism can guarantee better than $2/m$-MMS which is very interesting. On one hand, from theoretical point of view, I get the appeal of trying to bridge this gap, but given the explanation I gave above, I think the known $\alpha$-MMS algorithms are strictly better to use in any implementation. So beyond theoretical curiosity, I do not find other motivations for the given mechanism.
> >
> > Nevertheless, I must mention that I really like the fact you mentioned in your response that indeed your mechanism only asks for the highest valued item. I think this should be more highlighted in the paper. I find this the most important factor that makes your mechanism comparable with the $\alpha$-MMS allocation which asks the agents to report all the values. I think more discussions need to be added to the paper to make the result in better perspective with what is already known. I should also add to the strengths of the paper that having only the ordering of the goods as prediction is a very reasonable assumption. I increased my score.

---

### Official Review · Reviewer_dEcz · 2024-07-10

**Soundness:** 4
**Presentation:** 3
**Contribution:** 4
**Rating:** 8
**Confidence:** 4

**Summary:**

This submission studies the problem of approximating truthful mechanisms for the Maximin-Share allocation of individual goods whenever agents have incentives. Specifically, the authors design a learning-augmented algorithm for allocating goods to agents, given a prediction over the agents' ordinal preferences over goods. Like other work on learning-augmented algorithms, the goal is to take advantage of the prediction to get a better approximation when it is accurate, while still being robust to inaccurate predictions.

The authors give results for both the two-agent case and the n-agent case. Their results are based on a novel framework for designing allocation algorithms which they term "plant-then-steal". At a high level, the framework operates by first applying some allocation procedure (to be instantiated by the algorithm designer) which treats the predictions over agent preferences as correct in order to split the goods into sets of bundles (one per agent). In the second step, the framework uses the predictions to "plant" each player's favorite item in someone else's bundle. In the third step, the framework "steals back" each agent's favorite item according to their reported preferences.

For two players, the authors instantiate the plant-then-steal framework in order to get a 2-approximation whenever the predictions are correct (i.e. whenever they are "consistent"), and a worst-case "robustness" guarantee of m/2 when the predictions are arbitrarily wrong. (Here m is the number of items.) The authors also show that the performance of their instantiation degrades gracefully as a function of how inaccurate their predictions are, as measured by the Kendall tau distance between the predicted agent preferences and their actual preferences.

The authors also study the setting in which the algorithm designer is given access to predictions which don't necessarily take the form of agent preference orders. They show a lower bound on the trade-off between consistency/robustness, then provide mechanisms for allocating items in this setting using their plant-then-steal framework.

Beyond two players, the authors provide a 2-approximation whenever predictions are consistent, and obtain robustness guarantees of (m - n/2 - 1). Finally, the authors empirically evaluate several allocation schemes on two-player instances, and find that algorithms based off of their plant-then-steal framework perform well.

**Strengths:**

While the Maximin-Share allocation problem has been well-studied in the literature, the authors are the first to study the role of predictions in this problem, to the best of my knowledge. The introduction of predictions is well-motivated, as it is natural for the algorithm designer to have some guess about the preferences of the agents. Moreover, the authors present a very comprehensive set of results for this setting - the depth and breadth of results in this submission is impressive.

**Weaknesses:**

I have no major complaints. One relatively minor criticism is that the paper may be hard to read for someone who is not already familiar with the Maximin-Share allocation problem. For example, exactly what an agent report is is never clearly explained.

**Questions:**

n/a

**Limitations:**

The authors have adequately addressed the limitations of their work.

---

> ### Author Rebuttal · Authors · 2024-08-07
>
> “the paper may be hard to read for someone who is not already familiar with the Maximin-Share allocation problem. For example, exactly what an agent report is is never clearly explained”
>
> -- We will make an effort to improve readability in the camera-ready version of the paper, including a clearer explanation of what agents report. Specifically, we will discuss how mechanisms can be implemented by requiring agents to report their full valuation vectors or, alternatively, only their favorite item from a bundle of items (for mechanisms involving two agents) or multiple favorite items (for mechanisms involving n agents). This point will be clarified in the final version.

---

> > ### Comment · Reviewer_dEcz · 2024-08-08
> >
> > Thanks for your reply. After reading the other reviews and responses, I have decided to maintain my score.
> >
> > Moreover, I do not view the fact that most results in this submission are for the two-agent case as a substantial weakness, as the two-agent version of this problem is sufficiently well-motivated. While more substantial results for the n-agent case would of course be interesting, I think that it would be unfair to the authors to have them be on the hook to completely solve the n-player settj g, as (I believe) the two-agent results already clear the bar for NeurIPS.

---

### Official Review · Reviewer_EdC1 · 2024-07-14

**Soundness:** 3
**Presentation:** 3
**Contribution:** 2
**Rating:** 5
**Confidence:** 4

**Summary:**

This paper designs truthful algorithms for the fair allocation of indivisible goods in the learning-augmented framework. The algorithm is said to receive predictions about all agents' utilities for all goods (agents have additive utilities) or their ranking over the goods. The fairness notion studied is MMS (the maximin share). The main focus of the paper is on truthful mechanisms.

Most of the paper is focused on two agents. The idea of the proposed plant-and-steal framework is to use the predictions to initiate an allocation $A_1, A_2$; take the favorite good for agent 1 from $A_1$ and plant it in $A_2$ and vice versa; and, now with the utilities that agents report, let the agents steal their favorite item from the other bundle. This achieves a 2-approximate MMS (2-MMS in short) when predictions are completely correct (consistency), and $\lceil m/2 \rceil$-MMS when predictions are incorrect which matches what is achieved by the best truthful mechanism in the standard setting without predictions (robustness).

Next, the paper focuses on when predictions are agents' rankings for goods instead of the actual cardinal utilities. Using the round-robin mechanism in the plant-and-steal framework achieves the same guarantees (2-MMS for consistency) and $\lfloor m/2\rfloor$-MMS for robustness). When the predicted rankings have a Kendall tau distance of at most d, this mechanism achieves $2 \sqrt{d} + 6$-MMS which interpolates between constant and $m/2$ in the worst-case.

The paper ends with synthetic experiments with two agents.
In the appendix, the paper discusses mechanisms with $O(\log m /\epsilon)$ communication (previous ones had $\Omega(m)$) that achieve $2 + \epsilon$ robustness. The paper also generalizes the results to $n$ agents, achieving $2$-consistency and weaker robustness guarantees.

There are some other results in the appendix as well exploring pointwise tradeoffs in the Pareto frontier.

**Strengths:**

- The problem of designing learning augment algorithms for approximate MMS allocations is new.
- The results for the two agents are almost tight given the priorly known hardness results. The Kendall tau parameterized results show a nice trade-off.
- The paper claims and proofs seem correct to me (though, I have not checked most of the proofs in the appendix.)
- The plant-and-steal mechanism is simple.
- The paper is overall easy to understand. (The writing and organization could be improved considerably, more on this below.)

**Weaknesses:**

- The predictions are either the entire valuation matrix or all the rankings, which contain a lot of information.
- Most of the paper is focused on two agents. Even for two agents, I don’t think we learn the complete picture from the paper. I couldn’t find a lower bound for the Kendall tau parameterized result.
- While I like the algorithm's simplicity, I find it a relatively marginal improvement over the truthful mechanism of Amanatidis et al [7]. The technical novelty of the paper isn’t such that we say it’s a strength of the paper, in my opinion. The proofs and algorithms heavily use the priorly known results about truthful mechanisms and their MMS approximations Amanatidis et al [6, 7].
- The main body of the paper isn’t very well organized for a NeurIPS paper. The technical section starts on page 6. The experiment results (figures) are all in the appendix. Please reorganize, and instead of having two pages for “our results” allocate it to the technical section that speaks more formally about them.
- I couldn’t find the definition of “success rate” in the experiments. The y-label in Figure 2 is set to $\epsilon$ which should have been the subcaption of the subfigures. All in all, I couldn’t understand the results of the experiments and verify the takeaways.
- In the experiments, it’s unclear why a Mallows model was not used to generate rankings, which is a more justified statistical model to sample rankings. See e.g. https://proceedings.mlr.press/v97/tang19a.html for the Mallows model parameterized by the Kendall tau distance. I’m also not convinced much by the utility sampling procedure. Is having high-medium-low valuations necessary? How would the results change if it was just unit-sum valuations uniformly sampled from the Dirichlet(1,...,1) (random unit-sum vectors) or Gaussian perturbations of some ground truth valuation vector?


Minor points:
- The MMS guarantee of $m - 3n/2 - 1$ should have some conditions between $n$ and $m$ perhaps $m \ge 2n$ or similar. It’d be great to be more precise in the theorem/lemma statements.
- In appendix A, there is a “$8/m$” probability mentioned, a $1/2$ and a $1/4$, which I don’t see why would sum to $1$.
- line 58, “With probability $2$”, should it be $½$?
- line 50, “959 / 720 > 4/3$”, the reverse inequality is true
- Please consider renaming Theorem F.1 to Theorem 4.2 as one cannot just search for the proof of Theorem 4.2 easily, e.g. using the package “thm-restate”.
- You could save precious space by presenting algorithm 2 and mechanism 1 next to each other.
- line 122, one can “ran” -> run
- line 125, more “then” -> than
- table 1, $\hat{n}$ is undefined, please revise that part
- line 158, an $(1+\epsilon/2)$  -> change an to a. (I think a one is correct not an one.)
- lines 183 and 184, related “works” -> work
- line 291, thenB-RR-Plant-and-Steal -> add space between “then” and “B-RR-…”

**Questions:**

- In the experiments, what is the definition of success rate? Have you considered other utility sampling methods and if so, how did the results differ?
- Are there any matching lower bounds for the $\sqrt{d}$ Kendall-tau result?
- Have you thought about identical utilities? Can we achieve better guarantees assuming that agents have identical utilities?

**Limitations:**

This is a theory paper and I don't see immediate serious concerns. I would recommend the authors to discuss how much inefficiency their method can have in terms of social welfare for instance. Discussing the remaining gaps in the analysis is also helpful for the reader or follow up work.

---

> ### Author Rebuttal · Authors · 2024-08-07
>
> * "The predictions are either the entire valuation matrix or all the rankings, which contain a lot of information."
>
> -- The predictions used are:
> 1) Rankings, which we think are much more plausible than the exact valuations – it’s easier to predict that item A is more valuable than item B than to accurately predict their exact valuations. For this type of predictions, we can even handle inaccurate predictions, when the inaccuracy is bounded.
> 2)  Small space predictions. Here we use predictions of size O(log(m)/\epsilon) which are significantly smaller than the O(mㆍlog(m)) space needed to represent rankings, not to mention the much larger spaced needed to represent valuation functions..
> 3) For the n agent mechanisms, we use rankings as predictions, plus indicators indicating which are the “large” items, and not the entire valuation matrix, as stated in line 961, P. 28.
>
> In conclusion, we never need to use the entire valuation matrix as prediction, since we aim to use realistic and robust predictions. We thank the reviewer for raising our awareness that this is not emphasized enough, and will add a discussion to the final version of the paper to make sure this point comes across.
>
> * "Most of the paper is focused on two agents."
>
> -- Since strong impossibilities for truthful fair division already arise for two agents, this setting is well studied in the literature (see [1,2,3,4] for instance), as better understanding in this limited setting might be later generalized to multiagent settings. Indeed, we show that our mechanism can be generalized to $n$ players (while becoming much more involved). In this setting we get weaker, yet non-trivial, robustness guarantees.
>
> * "Are there any matching lower bounds for the 𝑑 Kendall-tau result?"
>
> -- We can show that the analysis of the O(\sqrt{d}) approximation is tight up to a constant for the mechanism at hand. We don’t give a general parameterized lower bound for every truthful mechanism. Showing general lower bound is highly non trivial for fair division and has been the focus of several papers on fair division, including specifically for approximate MMS allocation [2]. We leave this for future work.
>
> * “The main body of the paper isn’t very well organized for a NeurIPS paper…”
>
> -- Thank you for your suggestions, we’ll implement these changes in the final version.
>
> * "I couldn’t find the definition of “success rate” in the experiments. The y-label in Figure 2 is set to 𝜖 which should have been the subcaption of the subfigures."
>
> -- In L.350, we define our benchmark as “the percentage of instances where both players receive at least (1-𝜖) of their MMS values for different values of 𝜖,” which corresponds to the success rate. To improve clarity, we will explicitly mention this and add it to the subcaption.
>
> * "In the experiments, it’s unclear why a Mallows model was not used to generate rankings, which is a more justified statistical model to sample rankings. See e.g. .../tang19a.html for the Mallows model parameterized by the Kendall tau distance. I’m also not convinced much by the utility sampling procedure. Is having high-medium-low valuations necessary? How would the results change if it was just unit-sum valuations uniformly sampled from the Dirichlet(1,...,1) (random unit-sum vectors) or Gaussian perturbations of some ground truth valuation vector?"
>
> -- We thank the reviewer for the relevant reference. We will look into the proposed model and other studied models to generate ranking predictions.
>
> Regarding valuations, we observed that some sampling methods, such as I.I.D. sampling and the proposed random unit-sum vectors with an arbitrary balanced partition, perform quite well. However, we believe these do not represent most real-life instances. Therefore, we chose to use a relatively simple yet non-trivial model to generate valuations with three types of items: low, medium, and high. In this model, there are more low-valued items than medium-valued items, and more medium-valued items than high-valued items. We believe this phenomenon matches many real-life scenarios and illustrates the importance of the different components of our mechanisms.
>
>
> * "Have you thought about identical utilities? Can we achieve better guarantees assuming that agents have identical utilities?"
>
> -- That’s a very interesting question! For identical utilities, round robin allocations, or any turn-based picking mechanisms^, are ex-post Incentive compatible^^ (EPIC). In this case, when an agent picks an item, they should always pick the best available item, which is what is implemented if agents report their true valuations/ranking and the mechanism uses a turn-based mechanism to determine the allocation. Thus, we are able to get ⅔ approximation to the MMS for two agents without any predictions for this truthfulness notion. For more than two players, it follows from [3] that we can get EPIC mechanisms with approximation ratios that depend on the number of players, but not the number of items, truthfully. If we get indicators for the large items as predictions, even without rankings, then now we can get constant factor approximation EPIC mechanisms.
>
> ^ In turn-based picking mechanisms, each turn an agent picks a number of items from the set of remaining items, where the number of items is picked in advance.
> ^^ In Ex-post Incentive compatible (EPIC), bidding truthfully is a Nash Equilibrium.
>
> [1] Benjamin Plaut and Tim Roughgarden. Almost Envy-Freeness with General Valuations, SODA 2018.
> [2] G. Amanatidis, G. Birmpas, G. Christodoulou, and E. Markakis. Truthful allocation mechanisms without payments: Characterization and implications on fairness, EC 2017.
> [3] G. Amanatidis, G. Birmpas, and E. Markakis. On Truthful Mechanisms for Maximin Share Allocations, IJCAI 2016.
> [4] Biaoshuai Tao. On existence of truthful fair cake cutting mechanisms, EC 2024.

---

> > ### Comment · Reviewer_EdC1 · 2024-08-14
> >
> > Thank you for the detailed response. My apologies for the very late reply. I have read all the reviews and your responses. Also, I recently checked the appendix more thoroughly and looked at some of the proofs. I do appreciate the effort put into finding different trade-offs between robustness and consistency and the attempted generalizations to $n$ agents. The main body, the two technical sections 3 and 4, do not convey the depth explored. I still believe the main body should be revised significantly as a NeurIPS paper (see below).
> >
> > I share the concern of reviewer iNp4. The strong inapproximability of MMS shown for truthful mechanisms from prior work, do limit the significance of this study. Robustness guarantees of $m/2$-MMS or sometimes $m$-MMS are quite weak.
> >
> > I hold a different view than reviewer dEcz on the main result for two agents. I still think most of the "heavy-lifting" for results of two agents, e.g., characterizing truthful mechanisms including working with ordinal preferences, and the positive and negative results on MMS approximability, is done by prior work. Also, we do not get that close to a full picture for two agents in terms of the trade-offs between robustness and consistency --- as many papers do in the learning augmented framework. That said, I like the simple and nicely presented steal-and-plant mechanism.
> >
> > I am increasing my score, mainly for the results in the appendix.
> >
> > On re-organizing:
> > One idea is to move the experiments to the appendix --- all the plots which are in the appendix unfortunately, which shouldn't be the case. Instead of the long our results section, the technical sections can better describe the results in details. Perhaps include one of the more technically novel proofs (or a sketch of it) would better be included.

---

### Author Rebuttal · Authors · 2024-08-07

We thank all the reviewers for their thoughtful and thorough reviews. We will address all their comments and edit suggestions to improve the final manuscript. We address each reviewer’s comments/questions in the individual rebuttal sections below.

---

### Decision · Program_Chairs · 2024-09-25

**Decision:**

Accept (poster)

**Comment:**

Reviewers liked the novelty of the learning-augmented framework and the technical results (for two agents). Some reviewers are not completely satisfied with the robustness guarantee for general number of agents and the weak MMS bound under the learning-augmented framework (which sacrifices some privacy to obtain the prediction), while some other reviewers found the two agent case to be reasonably well-motivated especially given the large literature in MMS. Overall, the novelty outweighs the technical limitations. We hope the authors find the reviews helpful. Thanks for submitting to NeurIPS!